

# Seasonal cycle of sea surface temperature in the tropical Angolan upwelling system

Mareike Körner[1], Peter Brandt[1,2], Marcus Dengler[1]

[1]GEOMAR Helmholtz Centre for Ocean Research Kiel, Kiel, Germany

5  [2]Faculty of Mathematics and Natural Sciences, Kiel University, Kiel, Germany

*Correspondence to*: Mareike Körner (mkoerner@geomar.de)

**Abstract.** The Angolan shelf system represents a highly productive ecosystem. Throughout the year sea surface temperatures (SSTs) are cooler near the coast than further offshore. Lowest SSTs, the strongest cross-shore temperature gradient and maximum productivity occur in austral winter when seasonally prevailing upwelling favourable winds are weakest. Here, we

10  investigate the seasonal mixed layer heat budget to analyse atmospheric and oceanic causes for heat content variability. By using different satellite and in-situ data, we derive monthly estimates of surface heat fluxes, mean horizontal advection and local heat content change. We calculate the heat budgets for the near coastal and offshore regions separately to explore processes that lead to the observed differences. The results show that the net surface heat flux warms the coastal ocean stronger than further offshore thus acting to damp spatial SST differences. Mean horizontal heat advection is dominated by meridional

15  advection of warm water along the Angolan coast. However, its contribution to the heat budget is small. Ocean turbulence data suggests that the heat flux due to turbulent mixing across the base of the mixed layer is an important cooling term. This turbulent cooling that is strongest in shallow shelf regions is capable of explaining the observed negative cross-shore temperature gradient. The residuum of the mixed layer heat budget and uncertainties of budget terms are discussed.




## 1 Introduction

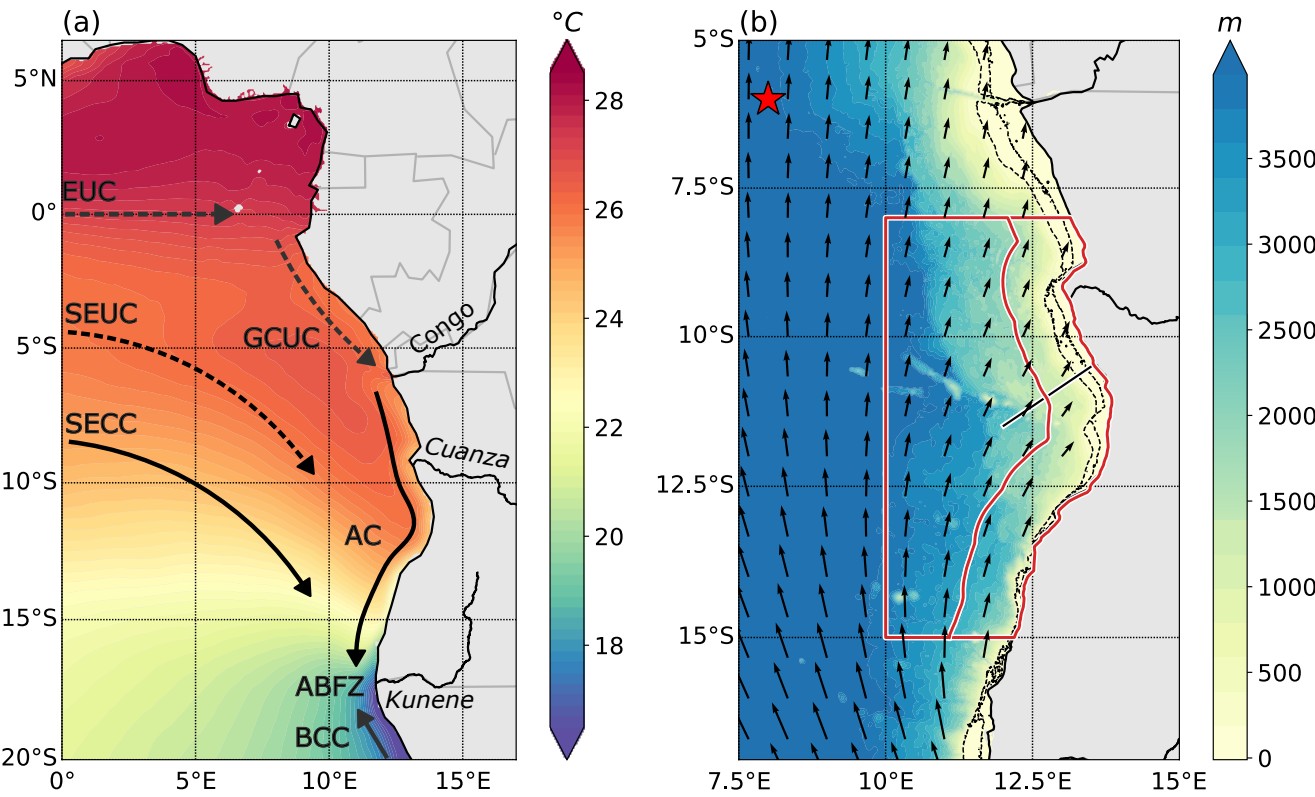

**Figure 1:** (a) Mean Sea Surface Temperature (colours) and schematic circulation in the south east tropical Atlantic. Solid arrows indicated pathways of surface currents, dashed arrows of subsurface currents. Currents displayed here are the Equatorial Undercurrent (EUC), the Gabon Congo Undercurrent (GCUC), the South Equatorial Undercurrent (SEUC), the South Equatorial Countercurrent (SECC), the Angola Current (AC), and the Benguela Coastal Current (BCC). Additionally, the position of the Angola-Benguela Frontal Zone (ABFZ) is marked. (b) Topography (colours) and mean wind field (arrows) off the coast of Southwest Africa. Black lines mark the 75 m and 175 m isobath. Red boxes show the extend of the coastal and the offshore box. Red star marks the position of the PIRATA-SEE mooring. Black line shows the 11°S section.

The coastal waters off Angola host a highly productive ecosystem of great socio-economic importance for local communities: the tropical Angolan Upwelling System (tAUS) (Sowman et al., 2010; FAO, 2011). The Congo River outflow at 6°S forms the northern boundary of the tAUS. To the south the Angola-Benguela Frontal Zone (ABFZ) located between 15°S and 18°S separates the warm surface waters of the tAUS from colder water further south (Fig. 1a). The tAUS is characterized by lower sea surface temperatures (SSTs) near the coast compared to further offshore throughout the year (Fig. 1a). This negative cross-shore SST gradient as well as the primary productivity peak in austral winter (Tchipalanga et al., 2018; Zeng et al., 2021; Awo et al., 2022). Thus, understanding the drivers of heat content changes in the upper ocean in the tAUS is important for the understanding of productivity. Additionally, it is also of global importance due to the remote impact of the southeast tropical Atlantic on tropical climate (Xu et al. 2014).



The circulation in the tAUS is dominated by the Angola Current (AC) (Fig. 1a) whose core is located at around 50 m depth. The AC transports warm water poleward along the Angolan continental slope and shelf. The transport is weak (~0.32 SV) and subject to variability on different time scales (Kopte et al., 2017). Past studies showed that the variability is connected to equatorial dynamics via propagation of equatorial and coastal trapped waves (CTWs) and to local forcings (Bachèlery et al., 2016; Kopte et al., 2017, 2018; Illig et al. 2018). The AC is fed via the Equatorial Undercurrent (EUC), the South Equatorial Undercurrent (SEUC), and the South Equatorial Countercurrent (SECC) with South Atlantic Central Water (SACW) (Kopte et al., 2017; Tchipalanga et al., 2018; Siegfried et al., 2019). In the ABFZ the poleward AC meets the northward Benguela Coastal Current (BCC) (Shannon et al., 1987; Siegfried et al., 2019; Fig 1a).

The surface waters in the tAUS are characterized by warm tropical conditions (Tchipalanga et al., 2018; Awo et al., 2022; Fig 1a). In austral winter the lowest SSTs are observed. The SST at the coast can then drop below 22.5°C. Highest temperatures are found in austral autumn when coastal SSTs can exceed 28°C (Awo et al., 2022). In contrast to other eastern boundary upwelling systems (EBUS) the changes in surface temperatures in the tAUS cannot be explained by local wind-driven upwelling (Ostrowski et al., 2009). On average, the winds in tAUS are southwesterly and substantially weaker than in the Benguela upwelling system (Fig. 1b). Its seasonal cycle is weak with a minimum in upwelling-favourable winds during the upwelling season in austral winter (Ostrowski et al., 2009; Zeng et al., 2021).

Changes in upper ocean heat content in the tAUS can be affected by the passage of remotely forced CTWs. The CTWs have a signal in sea level anomaly (SLA). Analysing SLA data in the tAUS reveals passage of four CTWs per year (Rouault 2012). In March a downwelling CTW propagate along the Angolan coast followed by an upwelling CTW in June-July. In October a second downwelling CTW arrives at the Angolan coast followed by a weak upwelling wave in December-January (Tchipalanga et al., 2018). Thus, the upwelling season coincides with the presence of an upwelling CTW at the Angolan coast. However, the vertical movement of the thermocline alone is unable to explain the near coastal cooling and the upward nutrient supply during austral winter. In this context the role of mixing induced by internal tides has been discussed (Ostrowski et al., 2009; Tchipalanga et al., 2018; Zeng et al., 2021). Zeng et al. (2021) showed in a recent model study that seasonal variations in the spatially-averaged generation, onshore flux, and dissipation of internal tide energy are weak. Due to the seasonal variation in stratification, however, diapycnal mixing driven by internal tides is more effective during the upwelling season.

The sea surface salinity (SSS) undergoes a distinct seasonal cycle in the tAUS (Awo et al., 2022). In October/November and in February/March freshwater intrude into the northern part of the tAUS (Kopte et al., 2017; Lübbecke et al., 2019; Awo et al., 2022). A recent model study (Awo et al., 2022) suggests that the freshening is controlled by meridional advection via the Angola Current where the Congo River is an important freshwater source. Vertical advection and mixing at the base of the ML was found to counteract this freshening (Awo et al., 2022).

The stratification in the tAUS is controlled by the passage of CTWs as well as the changes in surface salinity and temperature (Kopte et al. 2017; Tchipalanga et al., 2018). The stratification undergoes a semi-annual cycle with strong stratification near





the surface during the passage of the downwelling CTW and surface freshening in February/March and October/November
(Kopte et al. 2017; Tchipalanga et al., 2018, Awo et al. 2022).

The southeast tropical Atlantic is subject to a warm bias in SST in global climate models (Richter 2015; Kurian et al. 2021; Farneti et al. 2022). The reasons for the warm bias are still under debate. Some studies suggest that the origin of the bias lies in the representation of the atmosphere. Here excessive shortwave radiation due to a poor representation of clouds (Huang et al. 2007), an atmospheric moisture bias (Hourdin et al. 2015; Deppenmeier et al., 2020) or errors in the wind forcing (Voldoire
et al., 2019; Richter et al., 2020, Kurian et al., 2021) have been discussed. The role of the correct representation of ocean dynamics has also been suggested as the source of the bias (Xu et al. 2014). In this context Deppenmeier et al. (2020) show that enhancing turbulent vertical mixing in ocean models help reducing the error.

Previous studies investigated the mixed layer (ML) heat budget in the southeast Atlantic Ocean to identify atmospheric and oceanic drivers of heat content variability (Scannell and McPhaden, 2018; Foltz et al., 2019; Deppenmeier et al. 2020). Scannell
and McPhaden (2018) analyse the ML heat budget from moored observations at 6°S, 8°E. They found that surface heat fluxes and vertical turbulent entrainment primarily control the changes in SST. Foltz et al. (2019) examined the residuum between the heat content change and the surface heat fluxes. They attribute horizontal heat advection and turbulent cooling as the main contributor to this residuum. Their results reveal a large residuum in tAUS of $\sim 60$ W m$^{-2}$ increasing towards the coast. This suggests that in the near-coastal area other processes lead to the cooling of the ML than further offshore.

In the present study we analyse the atmospheric and oceanic drivers of heat content variability in the tAUS. In contrast to previous studies, we evaluate the ML heat budget near the coast and further offshore separately. This allows us to investigate and discuss processes that lead to the observed stronger cooling close to the coast. Furthermore, utilizing shipboard measurements of ocean turbulence we present for the first time an estimate of the impact of turbulent heat loss at the base of the ML in the tAUS. The study is structured as follows: In section 2 and 3 data and methodology are described, respectively.
In section 4 we present the results of our study and in section 5 we summarize and discuss the results.

## 2. Data

### 2.1 Shipboard measurements

In this study we analyse data collected during 6 research cruises that have been conducted in Angolan waters between 2013 and 2022 on board of RV Meteor. During those cruises ocean turbulence data was collected using a microstructure profiler
manufactured by Sea & Sun Technology. The microstructure shear measured by the microstructure profiler can be used to estimate the dissipation rate of turbulent kinetic energy (TKE). The microstructure profiler was equipped with 2-3 air foil shear sensors, an acceleration sensor, tilt sensors, a fast temperature sensor as well as standard CTD sensors. The microstructure profiles are measured by letting the loosely tethered probe fall free with a fall speed of 0.5 - 0.6 m s$^{-1}$.



During the 6 cruises, a total of 701 microstructure profiles were measured. The schedule of the cruises as well as the number

of microstructure profiles taken during the individual cruise are summarized in Table 1. A similar sampling strategy was chosen during the individual cruises that included a heavily sampled cross-shelf section at 11° S (Fig. 1b). However, the exact location of microstructure measurements on the shelf differed amongst the cruises, which leads to an inhomogeneous distribution of microstructure profiles in different months. The distribution for each cruise is displayed in Fig. 2.

| Time | Cruise ID | Number of microstructure profiles |
|------|-----------|-----------------------------------|
| July 2013 | M98 | 212 |
| October/November 2015 | M120 | 62 |
| October/November 2016 | M131 | 44 |
| June 2018 | M148 | 135 |
| September 2019 | M158 | 41 |
| April 2022 | M181 | 207 |

**Table 1:** Overview of the time and number of microstructure profiles measured during the 6 research cruises analysed in this

study.

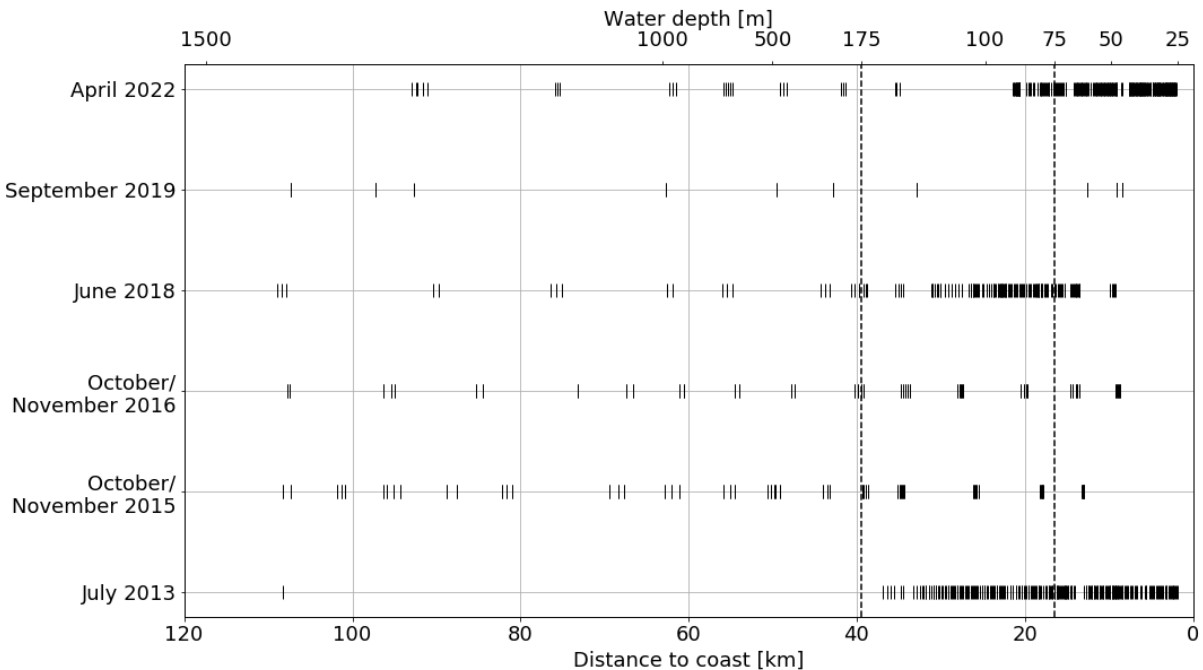





**Figure 2:** Distribution of microstructure profiles at the 11°S section as a function of distance to the coast, cruise and water depth. Each tick marks a microstructure profile. The vertical dotted lines mark the three areas analysed in section 4.3: shallow water (<75 m), shelf break area (> 75 m and < 175 m), and deep water (> 175 m).

**2.2 Mooring data**

We compare satellite data products of SST, near surface horizontal velocities, and surface heat fluxes to data measured by a mooring at 6°S, 8°E (Fig. 1). The mooring is the Southeast Extension (SEE) of the Prediction and Research Moored Array in the Tropical Atlantic (PIRATA) program. The PIRATA-SEE mooring was deployed for one year between June 2006 and June 2007. In June 2013 it was redeployed until September 2018. In March 2019 it was redeployed again for 6 months.

**2.3 Satellite and reanalysis data**

Different satellite and reanalysis products are used to estimate terms of the seasonal ML heat budget equation. As the datasets are available for different periods of time, we restrict our analysis to the time period between 1993 and 2018 for which all products mention below are available.

**2.3.1 Surface heat fluxes**

The climatologic net surface heat fluxes are derived from satellite data. Short- and longwave radiation are taken from the TropFlux product (Kumar et al., 2012). The data is available on a 1°x1° grid from 1979 to present at a daily and monthly resolution. However, at the time when this study was conducted, only data until December 2018 was made available.

Latent and sensible heat flux are taken from the MERRA2 product (GMAO, 2008). The monthly mean fields are available on a 0.5° longitude x 0.667 ° latitude grid from 1979 onward.

We made the choice to use different data products for the individual terms of the surface heat fluxes after comparing different data products to the surface fluxes measured by the PIRATA-SEE mooring (Appendix A).

**2.3.2 Sea surface temperature**

SST analysis are based on the OSTIA product (Good et al., 2020). The OISTA product uses satellite data as well as in situ measurements to provide global, daily, gap-filled SST fields. The data is available on a 0.05°x0.05° grid from 1981 onward.

**2.3.3 Surface velocities**

Estimates of horizontal heat advection are based on near-surface velocities of the OSCAR (Ocean Surface Current Analysis Real-time) product (ESR, 2009). The OSCAR dataset derives near surface ocean currents by using quasi-linear and steady flow momentum equations thus combining geostrophic, Ekman and Stommel shear dynamics. The basis is satellite and in situ measurements of sea surface height, surface vector wind and SST. The data is available on 1/3°x1/3° grid with a temporal
resolution of 5 days from 21 October, 1992 onward.





### 2.3.3 Mixed layer depth

ML depth (MLD) is taken from the PREFCLIM climatology (Rath et al., 2016). The climatology is based on monthly estimates of MLD at a 0.25°x0.25° resolution. The climatology contains all publicly available data sets from the World Ocean Database (MIMOC, Schmidtko et al., 2013). Additionally, hydrographic profiles from the EAF-Nansen program were considered
(Tchipalanga et al., 2018). The climatology uses an approach of Holte & Talley (2009) to determine the depth of the mixed layer. The climatology sets the minimum MLD to 10 m.

## 3. Methods

### 3.1 Mixed layer heat budget

To assess the oceanic and atmospheric driver of heat content changes we calculate the ML heat budget by following the
approach used by numerous previous observational studies (Stevenson & Niiler, 1983; Moisan & Niiler, 1998; Foltz et al., 2003, 2013; Hummels et al., 2014). The equation for the local heat balance in the ML can be expressed as:

$$h\,\rho c_p\,\frac{\partial T}{\partial t} = -\rho c_p\,h\,\vec{v}\,\cdot\,\vec{\nabla}T + q_{net} + r \qquad (1)$$

where $h$ is the ML depth, $c_p$ is specific heat capacity, $T$ the mean ML temperature, $\vec{v}$ the mean horizontal velocities in the ML, $q_{net}$ the net surface heat flux corrected for the part that penetrates through the ML, and $r$ is the residual. Changes in the local
heat content are balanced by the mean horizontal advection, the net surface heat flux, and the residual $r$. The residual contains errors of the other terms of Eq. 1 and other processes. One of these processes, on which we will focus in the present study, is the heat loss due to turbulent mixing across the base of the ML, termed turbulent heat loss in the following. The influence of this term will be discussed based on estimates of mixing strength utilizing microstructure data collected during 6 cruises in the tAUS. However, the available data is not extensive enough to calculate a seasonal cycle of the turbulent heat loss. Other
processes that are not evaluated here include the horizontal heat advection on temporal and spatial scales unresolved by the data used here (see section 3.1.2), vertical temperature velocity covariance and entrainment (Foltz et al., 2013; Stevenson & Niiler, 1983).

The evaluation of the terms of the ML heat budget is done using a box averaging strategy. For that we consider two boxes (Fig. 1b). The coastal box includes the area from 8° S to 15° S within 1° distance to the coast. The offshore box has the same
latitude range and extends from the coastal box to 10° E. All gridded terms are averaged spatially over the extend of the boxes. If a term of the ML budget consists of several variables with different spatial or temporal resolution, we interpolate the variable with the coarser resolution onto the higher resolution grid. A climatology of ML density and specific heat capacity is also calculated using the PREFCLIM climatology. Furthermore, all gridded terms are averaged over the same time period (1993-2018).



### 3.1.1 Surface heat fluxes

The net surface heat flux consists of the sum of longwave and shortwave radiation as well as the latent and sensible heat fluxes. Shortwave radiation is corrected for the amount of radiation that penetrates through the mixed layer while considering the absorption by phytoplankton. The vertical penetration of shortwave radiation can be estimated from climatological Chlorophyll-a concentrations. Morel & Antoine (1994) parameterize the irradiance at a certain depth applying three exponentials. The first describes the absorption of the infrared part of the sun spectrum depending on the angle of the incoming radiation. It is decaying on length scales between 0 to 0.267 m. The second and third exponential express the absorption of the longer- and shorter-wavelength part of the visible part of the spectrum. We find that only the third exponential is of interest for our application as the decaying scales of the first two exponential are much smaller than the ML depth. Thus, the fraction of shortwave radiation penetrating through the ML is:

$$\frac{E(-h)}{E(0)} \approx (1-R)V_2 \exp\left[-\frac{h}{Z_2}\right] \tag{2}$$

where $R$=0.43 is the infrared part of the sun spectrum and $V_2$ and $Z_2$ are polynomials of order 5 calculated with the monthly climatological Chlorophyll-a concentration and the constants given in Morel & Antoine (1994).

### 3.1.2 Mean horizontal heat advection

The mean horizontal heat advection is calculated using the OSCAR surface velocities and the horizontal gradient from the OISTA-SST product. Both the temporal and spatial resolution of the OSCAR surface velocities is coarser than of the OSTIA SST. The OSCAR surface velocities are available with a 5-day resolution on a 1/3°x1/3° grid. However, also the OSTIA product has limited effective resolution in the region of the Angolan upwelling system. Due to persistent cloud cover in the area, high-resolution passive infrared SST data are rarely available and the SST retrieval largely has to rely on low-resolution (50-60 km) passive microwave data (Nielsen-Englyst et al., 2021). Thus, using these datasets we are not able to resolve horizontal heat advection on temporal scales shorter than 5 days and spatial scales smaller than passive microwaves data resolution.

### 3.1.3 Turbulent heat loss at the base of the mixed layer

Turbulent heat fluxes are estimated from microstructure shear measurements. This method is explained in Hummels et al. (2014). A brief summary is provieded below.

Data from air foil shear sensors attached to microstructure probes are used us to estimate dissipation rates of turbulent kinetic energy, $\varepsilon$, via the variance method while assuming isotropy. Through integration the shear wave number spectrum, $E\ du/dz\ (k)$, is related to the dissipation rate of turbulent kinetic energy as



$$\varepsilon = 7.5\nu \int_{k_{min}}^{k_{max}} E_{du/dz}(k)dk \tag{3}$$

where $\nu$ is the dynamic viscosity of seawater. The shear spectra are calculated from overlapping two-second ensembles which
corresponds to ~ 1 m vertical resolution. Subsequently, spectra are integrated between a lower ($k_{min} = 3\ cpm$) and higher
wavenumber, $k_{max}$. The latter depends on the turbulence levels and the noise level. To account for variance loss due to the
limited resolution in wavenumber space, the spectra is fitted to the universal Nasmyth spectrum (Wolk et al.,2002).

The turbulent eddy diffusivity of mass is then calculated using $K_\rho = \Gamma\varepsilon N^{-2}$ (Osborn, 1980) where $\Gamma$ is the mixing efficiency
(set to 0.2 following Gregg et al., 2018) and $N^2$ is the buoyancy frequency squared, calculated from temperature, salinity, and
pressure data recorded by the microstructure profiler. $N^2$ profiles were smoothed to vertical scales larger than Ozmidov scale
by using a least square fitting method to vertical property gradients. The window size is chosen depending on the distance to
the ML with 3 m directly below the ML increasing linearly to 30 m. Finally, the turbulent heat flux is estimated from $J_H = -\rho c_p K_\rho\ \partial T/\partial z$.

Turbulent heat fluxes are calculated for each profile individually. For that we change the vertical coordinate to MLD + $\Delta$z with
a vertical resolution of 2 m. All measurements in the ML as well as 2 m below it are disregarded.  Note that we do not
interpolate the dissipation rates of TKE onto this grid but average all measurements in the respective depth bins. The binned
profiles of dissipation rates of TKE are then used to calculate the turbulent eddy diffusivity and finally the turbulent heat flux
for each profile.

**3.2 Uncertainty estimation**

The uncertainty of the monthly terms of the ML heat budget are calculated via: $err_{total} = \sqrt{err_{data}^2 + err_{seasonal}^2}$ . $err_{data}$
is the uncertainty arising from the data collection. To estimate the uncertainty of the satellite/reanalysis products for this region,
we calculate the RMS difference between the data and the data recorded by the PIRATA-SEE buoy. See Appendix A for the
comparison of the individual variables used in the study. The seasonal error ($err_{seasonal}$) arises from the fact that we use a
finite length of data record and is the standard error of each month. The error of the terms of the ML heat budget calculated by
combining different variables are calculated using standard error propagation.

To evaluate the uncertainty connected to the turbulent mixing we use the method of bootstrapping following the approach of
Hummels et al. (2014). This method gives us the 95% confidence levels.



## 4 Results



**Figure 3:** (a), (b), (d), (e) Seasonal mean sea surface temperature (colours) and surface velocities (arrows). Arrow length is referenced in the lower right corner of each subplot. Black boxes show the coastal and the offshore box used for calculating the mixed layer heat budget. (c) and (f) Hovmoeller plots of MLD as a function of latitude and month zonally averaged over the (c) offshore and (f) coastal box.

Although the tAUS region is situated in the tropics, SSTs undergo an elevated seasonal cycle (Fig. 3 a,b,d,e). The highest temperatures are found during austral autumn, reaching their maximum in March (28.1 °C averaged over the coastal and 28.2 °C averaged over the offshore box). The lowest temperatures are observed during austral winter in August (20.9 °C in the coastal and 21.5 °C in the offshore box). Accordingly, the ML cools between March and August and warms during the rest of




the year. In the following sections we analyse the atmospheric and oceanic processes that impact the described heat content changes followed by an analyses of the resulting ML budget.

Before turning to the individual processes, we look at the MLD and its changes throughout the year (Fig. 3 c, f). In general, the ML is shallower at the coast than further offshore. Additionally, the ML deepens with increasing latitude. In both boxes the deepest ML is found in August when the average MLD is 14.2 m in the coastal and 17.6 m in the offshore box. The shallowest ML is present in February when it averages 12.7 m in the coastal box and 14.1 m in the offshore box.

## 4.1 Surface heat fluxes

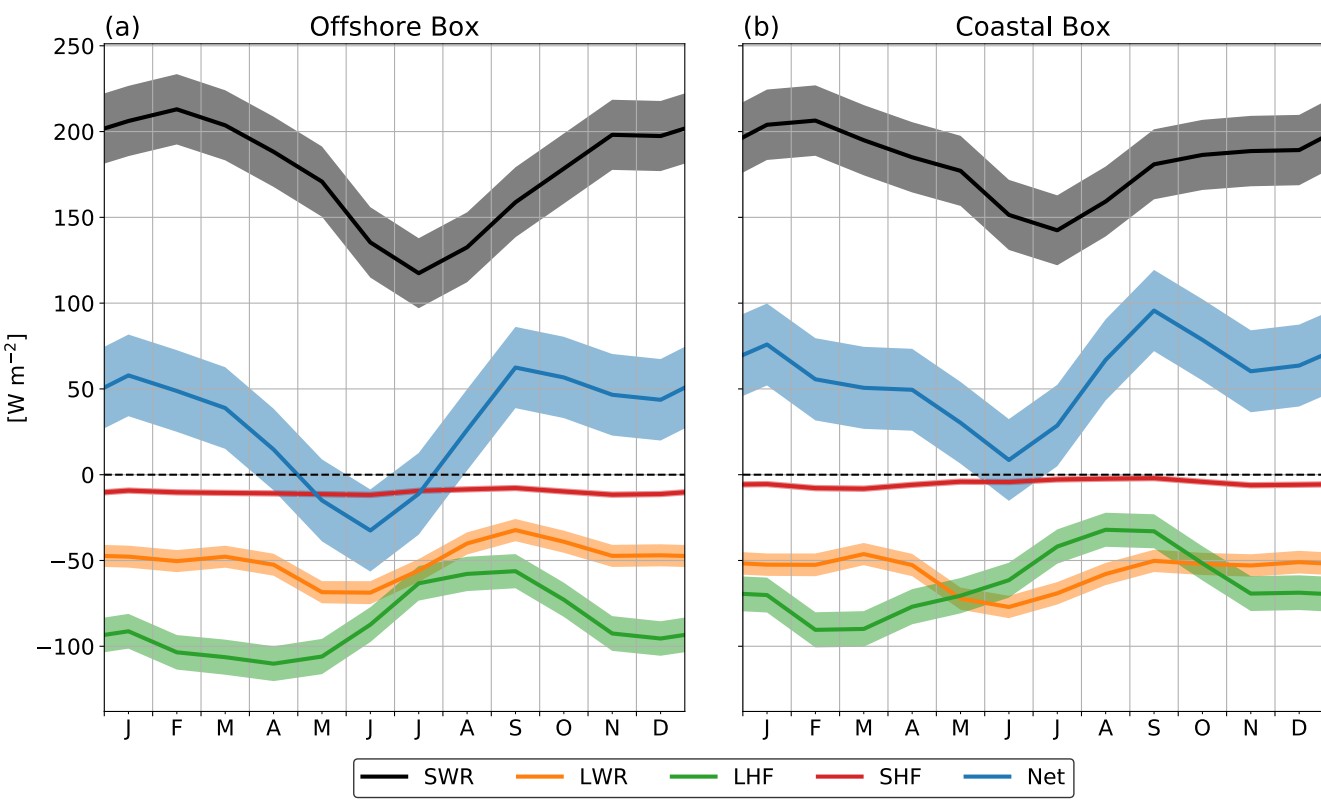

**Figure 4:** Climatology of surface heat fluxes averaged over the (a) offshore and (b) coastal box. Black line shows the climatology of the shortwave radiation (SWR), orange line of the longwave radiation (LWR), green line of the latent heat flux (LHF), red line of the sensible heat flux (SHF), and blue line of the net surface heat flux. Shaded areas give the uncertainty estimate of the respective fluxes (see Sect. 3.2).

The surface heat fluxes show distinct seasonal cycles having similar characteristics in the offshore and the coastal box (Fig. 4). Differences between the boxes lie foremost in the amplitude and strength of the respective seasonal cycles. In both boxes shortwave radiation peaks in February. The minimum is found in July driven by the seasonal maximum in solar zenith angle and the expansion of the low cloud cover (Scannell and McPhaden, 2018). The average shortwave radiation is slightly higher





in the coastal box. Largest differences are observed in austral winter when shortwave radiation is higher near the coast
(difference of 25 W m$^{-2}$ in July). The shortwave radiation is corrected for the amount of radiation that penetrates through the
base of the mixed layer. This drives the larger differences in austral winter between both areas as a higher concentration of
Chlorophyll-a near the coast leads to more absorption of shortwave radiation by the mixed layer. The longwave radiation has
its largest cooling effect in June. Between June and October, cooling by the longwave radiation is stronger in the coastal box.
The latent heat flux has a similar seasonal cycle in both boxes. The smallest cooling effect is found between June and September
when wind speeds are at their seasonal minimum. Latent heat flux cools the ML the strongest between February and May. The
increasing wind speed away from the coast (Fig. 1b) lead to an overall stronger cooling in the offshore box. The sensible heat
flux is small in both boxes and constitutes a minor contribution to the net surface heat flux.

The resulting net surface heat flux has its minimum in June and its maximum in September in both boxes. The differences in
the individual surface heat flux terms result in a stronger net surface heat flux in the coastal box compared to the offshore box.
Thus, the net surface heat flux actually acts to damp the observed SST differences between the coastal and the offshore area.
Consequently, the surface fluxes are not able to explain the signal of cold water in the near coastal area of the tAUS. Note that
the differences between the offshore and coastal box peaks between May and August when it is ~40 W m$^{-2}$ stronger in coastal
box.

## 4.2 Mean horizontal advection

The seasonal cycle of the mean horizontal heat advection is determined by the seasonal cycle of the horizontal temperature
gradient and the surface velocities (Eq. 1). Fig. 3 shows that within the coastal box the temperature decrease towards the coast
throughout the year. This negative zonal temperature gradient is strongest between May and August (~ 12 x 10$^{-3}$ °C km$^{-1}$). A
secondary maximum is found in January. In contrast, the meridional temperature gradient within the coastal box is always
positive as SSTs increase towards the equator. On average its magnitude is 5 x 10$^{-3}$ °C km$^{-1}$ while its seasonal cycle is weak.
The meridional temperature gradient averaged over the offshore box is of similar strength (on average 5 x 10$^{-3}$ °C km$^{-1}$) and
also exhibits a weak seasonal cycle. The offshore zonal temperature gradient is always positive as well (on average 3 x 10$^{-3}$
°C km$^{-1}$).

The velocity field off the coast of Angola is in generally weak (Fig. 3). Close to the coast, velocities along the coast dominate.
Here, the velocities in the northern part of the tAUS are elevated compared to further south throughout the year. The southward
velocity component peaks in October (9 cm s$^{-1}$ averaged over the coastal box). Note that this maximum agrees well with the
seasonal maximum of southward velocities of the Angola Current as shown from moored velocity observations in Kopte et al.
(2017). A secondary southward velocity maximum is found in February. The weakest meridional velocities are found in August
when velocities are close to zero. In the offshore box the velocity field is weaker and noisier than in the coastal box. One
feature present throughout the year seems to be an anticyclonic rotation centred around 12° S, 12° E. Averaged over the





offshore box the surface velocities do not exceed 3 cm s⁻¹ throughout the year. Furthermore, annual averaged velocities are smaller than 1 cm s⁻¹.

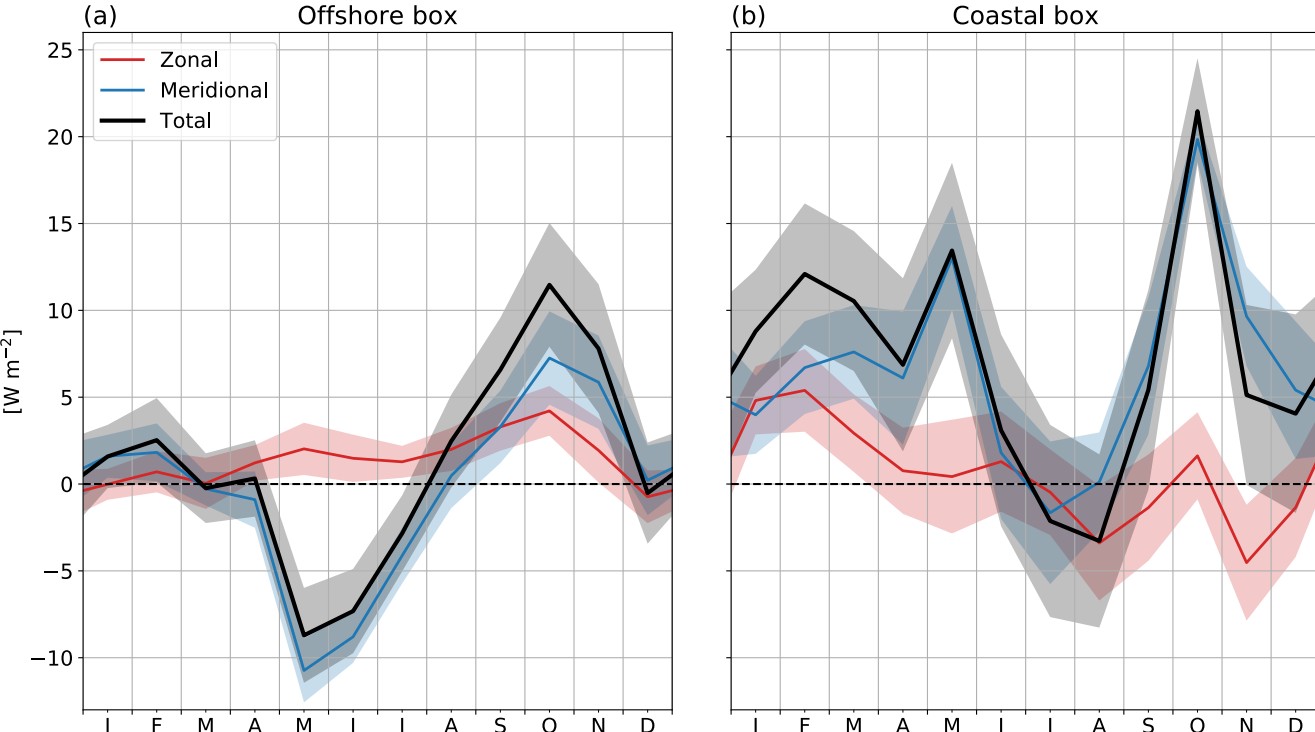

**Figure 5:** Seasonal cycle of mean horizontal heat advection averaged over the coastal box (a) and the offshore box (b). Red lines present the mean zonal horizontal heat advection, clue lines the mean meridional horizontal heat advection, and the black
lines the sum of both. Shaded areas give the estimated error (see Sect. 3.2).

The resulting mean zonal and meridional heat advection are presented in Fig. 5. In both boxes, the total mean horizontal heat advection is dominated by the meridional component. Averaged over the year, the mean horizontal heat advection warms the ML in both regions, but its contribution is small compared to the net surface fluxes. The maximum in both boxes is reached in October when southward velocities are at the seasonal maximum. Then, horizontal heat advection amounts $21.5 \pm 3$ W m⁻²
when averaged in the coastal box and $11.5 \pm 3$ W m⁻² for the offshore box. Note that mean horizontal heat advection are calculated using five-day velocities available on a 1/3° grid. Heat advection on shorter time scales and smaller spatial scales cannot be determined from currently available datasets. This will be discussed in Sec. 5.

## 4.3 Turbulent heat loss at the base of the mixed layer

As has been reported from other upwelling regions (e.g., Perlin et al., 2005; Schafstall et al., 2010), the microstructure profiles
available to this study (section 2.1) indicates a strong dependence of the TKE dissipation rate on bathymetry. This is illustrated




in Fig. 6 showing the mean distribution of TKE dissipation rates across the continental slope and shelf from 6 cruises at 11° S (see Fig. 1b and Fig. 2 for details on data coverage).

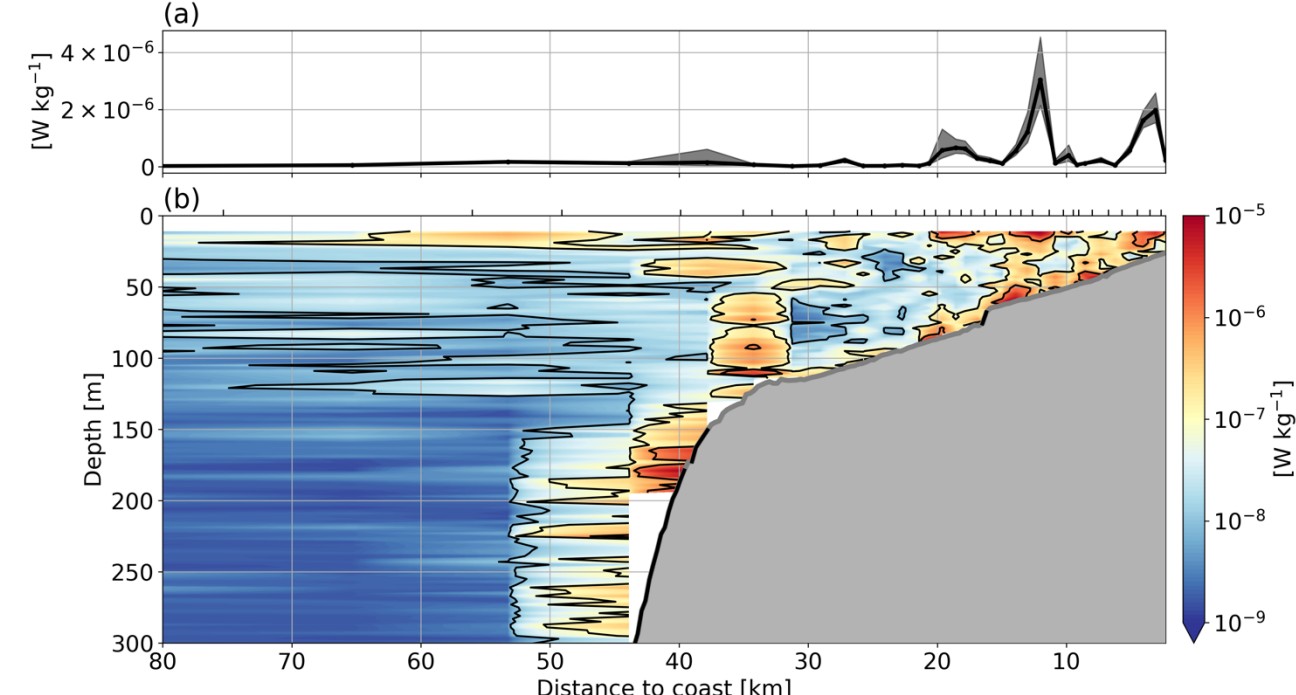

**Figure 6:** Dissipation of TKE at the 11°S section (see Fig. 1b) as a function of distance to coast [km]. Microstructure data are binned together in groups of 20 profiles. (a) Mean dissipation of TKE averaged between 2 and 15 m below the ML. Grey shading shows 95% confidence interval calculated via bootstrapping. (b) Section of mean dissipation of TKE against depth and distance to coast. Topography coloured in black marks the supercritical slope for the M2 tide calculated with the time averaged 11°S stratification from Kopte et al. (2017). Black ticks at the plot mark the borders of the 20-profile groups.

Elevated dissipation rates of TKE close at the surface as well as in and above the bottom boundary layer are revealed. Furthermore, dissipation rates above $10^{-7}$ W kg$^{-1}$ are found in the whole water column at the shelf break and in waters shallower than 75 m. In the depth range between 2 m and 15 m below the ML (Fig. 6 a), which is relevant for determining the turbulent heat loss at the base of the ML, a TKE dissipation rate dependence on the water depth is also evident. Here, high mean dissipation rates of TKE that can exceed 1 x $10^{-6}$ W kg$^{-1}$ are particularly frequent in waters shallower than 75 m. Note that the microstructure shear data were taken during different seasons. However, we find similar dependences of dissipation rates of TKE on water depth when considering data from individual cruises separately (not shown). Thus, the cross-slope distribution of TKE dissipation rates likely does not exhibit elevated seasonal variability.



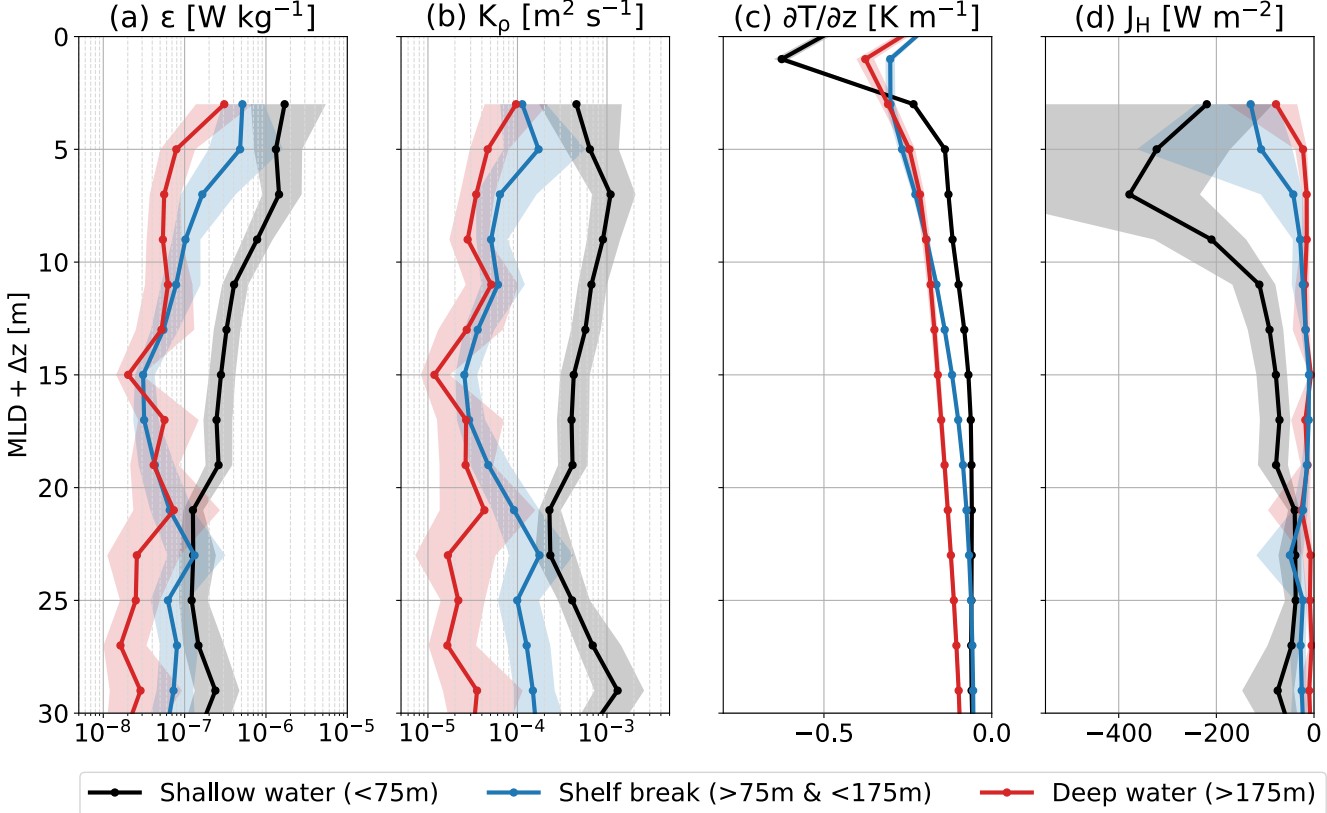

**Figure 7:** Averaged profiles as a function of distance to the ML. Profiles taken during 6 different cruises are allocated into three groups according to the water depth where the profile was taken. Profiles taken in shallow water ($< 75$ m, black), in the area of the shelf break ($>75$ m and $<175$ m, blue), and in deep waters ($>175$ m, red) are grouped together. (a) shows the dissipation rate of TKE [W kg$^{-1}$], (b) the eddy diffusivity [m$^2$ s$^{-1}$], (c) the vertical temperature gradient [K m$^{-1}$], and (d) the turbulent heat flux [W m$^{-2}$]. The shaded areas give the 95% confidence intervals.

We conclude from the mean distribution of dissipation of TKE that turbulent heat flux at the Angolan shelf has to be analysed dependent on the water depth of the respective microstructure profile. The 701 microstructure profiles were thus allocated in three groups based on water depth: profiles measured in water deeper than 175 m (deep water), profiles measured in water depth between 75 m and 175 m (shelf break area), and profiles taken in water depth shallower than 75 m (shallow water). Individual profiles were mapped as a function of vertical distance to the ML in 2 m bins prior to averaging (Fig. 7).

The results in Fig. 7 clearly shows differences between the three regions. The highest dissipation rates of TKE below the ML are found in shallow waters. These elevated dissipation rates of TKE ultimately lead to strongly elevated turbulent heat fluxes. Averaged between 2 and 15 m, the heat flux is -188 [-159, -222] W m$^{-2}$ in shallow waters (Table 2). In the same depth range the shelf break area exhibit -49 [-42, -58] W m$^{-2}$. The heat loss in deep waters is even smaller (-24 [-21, -29] W m$^{-2}$). These results show that turbulent heat loss at the base of the ML is an important cooling term of the ML heat budget. Here, especially the shallow waters play an important role as the heat loss is elevated by about a factor 8 compared to deep waters.





Contrary to the shelf break area and deep waters, the maximum turbulent heat flux in shallow waters is not found directly below the ML but 7 m below it (Fig. 7 d). Note that in shallow waters the dissipation rates between 2 m and 7 m below the ML are of similar order. Thus, different strength of dissipation rates cannot explain the maximum at 7 m alone. Additionally, we have to analyse the stratification. The stratification, similarly to the vertical temperature gradient (Fig 7c), is strong just below the ML decreasing rapidly with increasing distance to the ML. The strong stratification decreases the eddy diffusivity

in contrast to region with a weaker stratification. It results in a maximum in eddy diffusivity at 7 m below the ML (Fig. 7b). This maximum in eddy diffusivity ultimately contribute to the maximum in turbulent heat flux at this depth and underlines the role of stratification for the turbulent heat flux. Note that the reason for the strong stratification below the ML as well as the consequences of the heat flux maximum not directly below the ML will be discussed in Sec. 5.

Until now we only discussed the turbulent heat flux as a function of bathymetry. For the analyses of the heat content change

throughout the year, the seasonality of the turbulent heat flux is also of interest. It is ambitious to discuss seasonal differences of turbulent heat flux based on the cruise data. The sampling strategy during the cruises was not the same leading to a different distribution of measured profiles along the 11°S section during the different cruises (Fig. 2, Table 2). To discuss temporal variability, we present the averaged turbulent heat fluxes in the three different depth ranges between 2 and 15 m during the different cruises (Table 2). The reported values clearly show a large variability. In shallow waters the fluxes range from -3 [-

2, -3] W m$^{-2}$(October/November 2015) to -390 [-326, -470] W m$^{-2}$(April 2022). Similarly, in the area of the shelf break fluxes range from -2 [-1, -2] W m$^{-2}$ (October/November 2015) to -135 [-116, -163] W m$^{-2}$(April 2022). In deep waters the minimum fluxes were measured during July 2013 (-1 [0, -1]) W m$^{-2}$) and the highest were measured during September 2019 (-46 [-34, -64] W m$^{-2}$). Note that two cruises conducted in the same month one year apart shows very different heat fluxes. In October/November 2015 an averaged flux of -3 [-2, -3] W m$^{-2}$ measured in shallow waters whereas one year later the average

flux is -232 [-188, -290] W m$^{-2}$. Because of this large variability, we abstain from including seasonal estimates of turbulent heat flux at the base of the ML in the budget. A possible seasonality in the turbulent heat flux term is discussed in Sec. 5. Furthermore, the calculated fluxes from the microstructure data can exhibits very high values. Especially the data recorded in April 2022 in shallow waters exhibit much higher heat losses than the amount of net surface heat fluxes that is put into the ML. In Sec. 5 we discuss possible explanation for these high heat losses.

| Time | Cruise | Shallow water (<75m) | | Shelf break area (>75m & <175m) | | Deep water (>175m) | |
|------|--------|----------------------|---|--------------------------------|---|--------------------|---|
| | | $J_H$ [W m$^{-2}$] | N | $J_H$ [W m$^{-2}$] | N | $J_H$ [W m$^{-2}$] | N |
| July 2013 | M98 | -27 (-23, -32) | 106 | -15 (-13, -17) | 105 | -1 (0, -1) | 1 |



| | | | | | | | |
|---|---|---|---|---|---|---|---|
| October/ November 2015 | M120 | -3 (-2, -3) | 5 | -2 (-1, -2) | 17 | -7 (-6, -10) | 40 |
| October/ November 2016 | M131 | -232 (-188, -290) | 10 | -85 (-70, -105) | 14 | -21 (-17, -25) | 20 |
| June 2018 | M148 | -86 (-71, -107) | 29 | -60 (-50, -75) | 84 | -36 (-30, -45) | 22 |
| September 2019 | M158 | -47 (-35, -66) | 16 | -17 (-13, -25) | 4 | -46 (-34, -64) | 21 |
| April 2022 | M181 | -390 (-326, -470) | 144 | -135 (-116, -163) | 38 | -26 (-22, -31) | 25 |
| **Mean** | | **-188 (-159, -222)** | **310** | **-49 (-42, -58)** | **262** | **-24 (-21, 29)** | **129** |

**Table 2:** Turbulent heat flux ($J_H$) averaged between 2 and 15 m below the ML during the respective cruise for profiles taken in different depth ranges. 95% confidence interval is given in the brackets below. The number of profiles in each depth range is presented as well (N).

## 4.4 Mixed layer heat budget

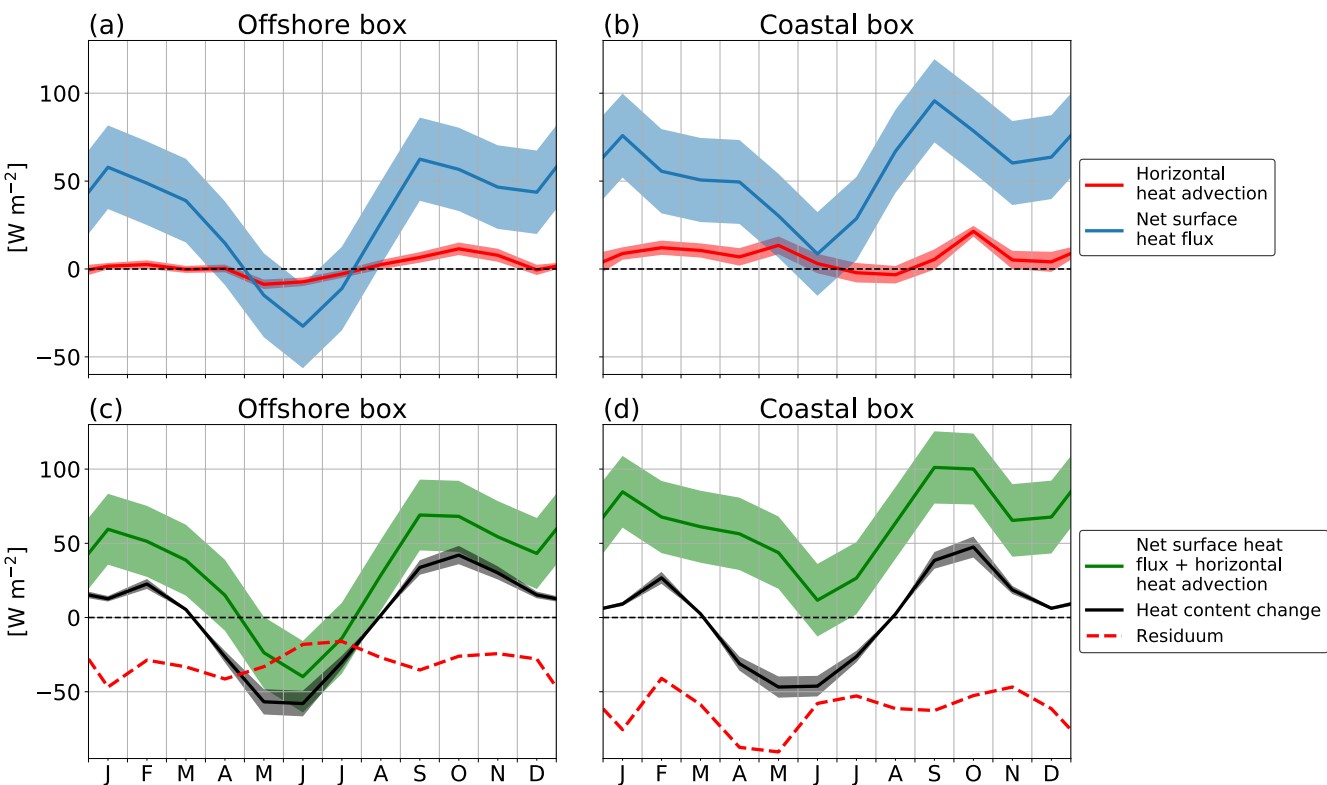






**Figure 8:** (a) & (b) Individual contributions to the ML heat budget. Colours are explained in the legend. (c) & (d) Sum of net surface heat flux and horizontal heat advection (green lines), the observed heat content change (black lines), and the resulting residuum between both (red dashed line). (a) & (c) show the result averaged over the offshore box, (b) & (d) display the results averaged over the coastal box.

Fig. 8 presents the individual terms of the ML heat budget analysed in this study as well as the resulting budget itself averaged over the coastal and the offshore box respectively. The net surface heat flux is an important term of the ML heat budget in both the offshore and the coastal box. In contrast, the mean horizontal heat advection term is small and of minor importance. Averaged over the year both terms have a warming effect on the ML in both boxes.

The sum of surface heat fluxes and mean horizontal heat advection show a similar seasonal cycle with different magnitudes in
the coastal and the offshore box (Fig. 8 c, d). It is characterized by the seasonal cycle of the net surface heat fluxes and only slightly modulated by the mean horizontal heat advection term. Consequently, the sum of both is positive throughout the year in the coastal box. Its maximum is found in September ($99 \pm 24$ W m$^{-2}$) while its minimum is found in June ($10 \pm 24$ W m$^{-2}$). In the offshore box the sum of net surface heat flux and total horizontal heat advection is negative between May and June. Its maximum is found in September ($68 \pm 24$ W m$^{-2}$) its minimum is detected in June ($-40 \pm 24$ W m$^{-2}$). The heat storage term
shows that the ML cools from March to August and warms in the rest of the year in both boxes.

Comparing the heat storage term with the sum of mean horizontal heat advection and net surface heat fluxes reveals that a large residuum remains in the coastal box as well as in the offshore box (Fig. 8 c, d). In the coastal box the residuum is considerably larger (on average 62 W m$^{-2}$). The average residuum in the offshore box is only half the size (30 W m$^{-2}$). The residuum undergoes a weak seasonal cycle which differs between the boxes.

The residuum includes, amongst other things, contributions of the turbulent heat flux at the base of the ML. While we cannot calculate a seasonal cycle of this term for the coastal and the offshore box based on the microstructure data, an average contribution for the coastal box can be estimated. Analysis of the microstructure profiles revealed a dependence of the turbulence heat flux on bathymetry. Thus, we consider a weighted mean based on the area of the coastal box that falls into the respective depth ranges discussed in Sec. 4.3. In total, the water depth in 12% of the coastal box area is shallower than 75 m,
water depth in 13% of the coastal box is between 75 and 175 m, and in 75 % it is deeper than 175 m. The resulting weighted mean calculated over all microstructure profiles averaged between 2 m and 15 m yields a contribution of -48 [-43, -55] W m$^{-2}$ to the ML heat budget. Comparing this to the average residuum of $62 \pm 20$ W m$^{-2}$ in the coastal box underlines that turbulent heat loss at the base of the ML is an important process contributing to the cooling of the ML in the tAUS. The microstructure profiles further suggests that this process is particularly important in near coastal areas as the turbulent heat flux is much larger here than further offshore. The larger residuum in the coastal box compared to the offshore box supports these results.

The residuum also includes biases in the evaluated terms of the ML heat budget (see Sec. 3.1). To estimate possible sources of biases, we compared the satellite/ reanalyses data to in-situ data measured at the PIRATA-SEE mooring site at 6°S, 8°E (see Fig. 1 for location). The comparison detailed in Appendix A revealed large differences of monthly-averaged surface heat





flux components despite being estimated over the same time span. In particular, the monthly averages of shortwave radiation
showed elevated differences between the TropFlux climatology and buoy shortwave radiation sensor data, suggesting that
TropFlux shortwave radiation is biased high. The differences of the net surface heat flux range between 2 W m$^{-2}$ in May and
38 W m$^{-2}$ in January. This suggests that the satellite/reanalyses data may overestimate the amount of net surface heat fluxes
and thus contribute to a positive residuum in the ML heat budget.

## 5 Summary & Discussion

The tAUS is a highly productive ecosystem. In the tAUS surface temperatures are lower near the coast compared to further
offshore. In austral winter, we find the lowest SSTs and the strongest cross-shore SST gradient in the tAUS. In this study we
calculate different terms of the ML heat budget based on satellite and reanalysis data to analyse atmospheric and oceanic
drivers of heat content variability. The heat budget terms are averaged over two boxes: One located directly at the coast of the
tAUS and one offshore of it. This allows us to analyse and discuss processes that might be of different importance in both
regions. Additionally, we analyse the impact of turbulent heat flux at the base of the ML based on shipboard observations
taken almost exclusively in the coastal box.

The surface heat fluxes are an important driver of ML heat content changes in the tAUS. The seasonal cycles of the heat flux
terms are similar near the coast and further offshore. The strongest cooling term is the latent heat flux, which is larger in the
offshore area of the tAUS due to decreasing wind speeds towards the coast. As warming due to shortwave radiation is elevated
in the coastal region as well, resulting net surface heat flux is larger in the coastal box compared to the offshore area. Thus,
net surface heat fluxes act to damp the observed cross-shore temperature gradient. Note that the differences are particularly
large during austral summer when the cross-shore temperature gradient is at its seasonal maximum (Fig. 9).

Mean horizontal heat advection contributes to warming of the ML in the coastal region as well as offshore. However, the term
is small, averaging only 7 W m$^{-2}$ (1 W m$^{-2}$) in the coastal (offshore) box. It is sustained by the southward advection of warm
equatorial waters by the Angola Current, which peaks during October.

The turbulent heat flux at the base of the mixed layer is estimated from shipboard microstructure measurements taken during
6 cruises. We find the amount of heat flux to vary with bathymetry. Highest TKE dissipation rates and consequently elevated
turbulent heat fluxes are found in waters shallower than 75 m, suggesting stronger cooling close to coast compared to further
offshore. This term thus acts to enhance a cross-shelf temperature gradient.

The net surface heat fluxes and horizontal heat advection are not able to explain the observed heat content changes. Our
analyses show that the turbulent heat flux at the base of the ML is able to explain a large proportion of the resulting residuum
in the coastal box. The averaged residuum in the coastal box is twice as large than that in the offshore box, which supports the
hypothesis that turbulent heat fluxes are more important in the proximity of the coast compared to further offshore.
Additionally, biases in the TropFlux surface heat fluxes may contribute to the residuum. As shown in the appendix, shortwave



radiation from the climatology is larger than that measured by in-situ sensors on a PIRATA buoy situated in proximity of the
       study site.

       Our analysis of the ocean turbulence data reveals a connection between the amount of turbulent heat flux and the bathymetry.
       Processes that lead to increased dissipation rates of TKE and ultimately increased turbulent heat fluxes in shallow waters on
       the Angolan shelf include internal waves and their interaction with the topography. Internal tides are assumed to be the largest
contributor to the internal wave energy on the Angolan shelf (Zeng et al., 2021). They are generated by the interaction of the
       barotropic tide and the continental slope. In the tAUS, the topography is critical or supercritical with respect to the M2 tide at
       the continental slope mostly in the depth range between 200 and 500 m (Fig. 6, Zeng et al., 2021). Here, the largest portion of
       the internal tide energy is generated. While part of that energy was found to be dissipated locally or further offshore, a
       substantial part propagates onshore and was found to be dissipated in shallow waters near the coast (Zeng et al. 2021). Note
that also smaller topographic features with critical slopes exist further onshore which can shape the local distribution of
       dissipation rates of TKE on the shelf as near-critical slopes are areas of enhanced velocity shear (Legg and Adcroft, 2003).

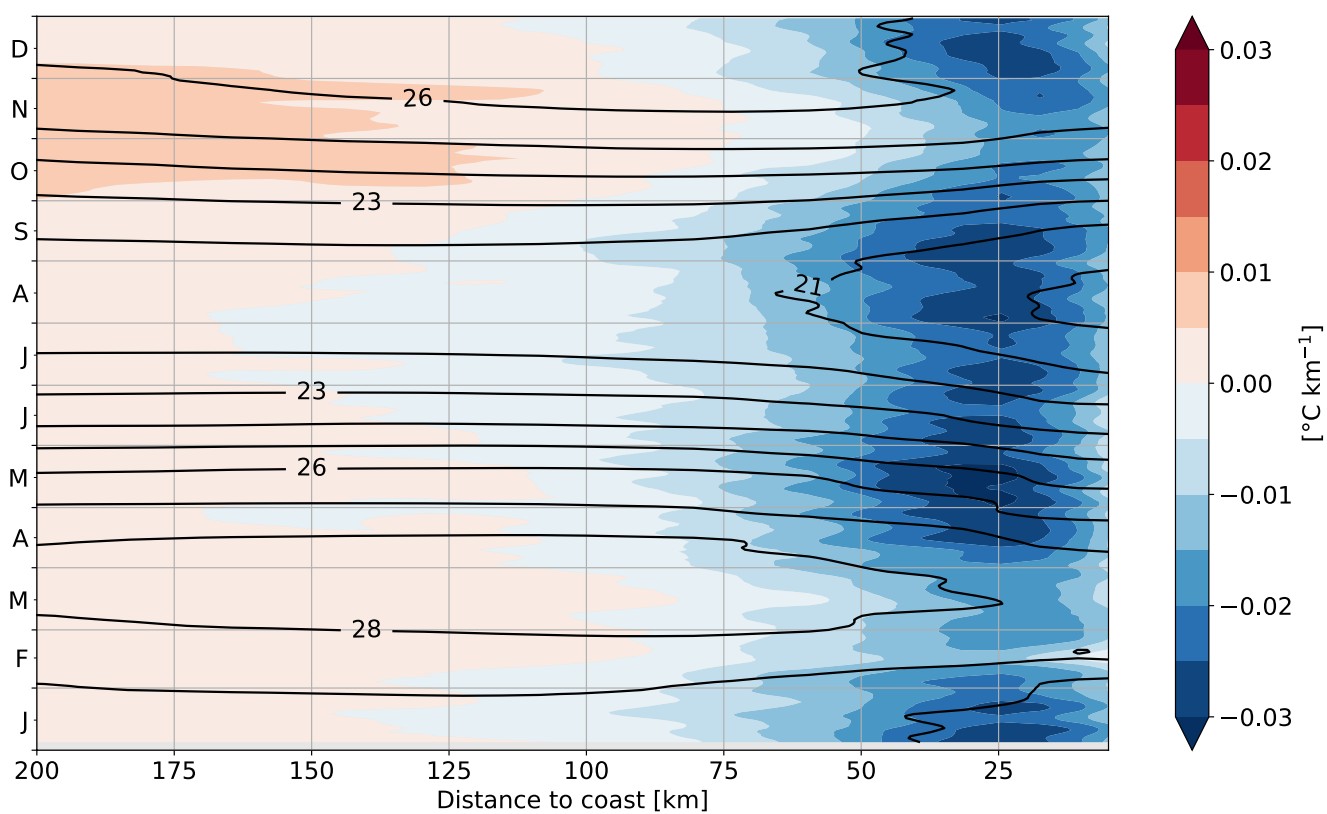

**Figure 9:** Zonal temperature gradient (colours) and SST (contours) as a function of distance to coast averaged between 8°S
and 15°S. Contour lines are every 1°C from 20°C to 28°C. The data are treated with a 5-day running mean.





For the seasonal ML heat budget, the seasonality of turbulent heat flux at the base of the ML is of interest. The turbulent heat flux calculated from microstructure profiles exhibit a high variability but also suggests that the turbulent heat loss is an important cooling term throughout the year. However, the data only provide snapshots of the dissipation at the Angolan shelf. A robust discussion of seasonal differences based on the data is thus ambitious. The model study of Zeng et al. (2021) showed that seasonal variations in the spatially averaged generation, onshore flux, and dissipation of internal tide energy are weak.

However, due to the seasonal variation in stratification (passage of CTWs, seasonal cycle of SSS, SST differences through surface fluxes) the mixing due to internal tides is more effective during austral winter. This result fits well to describe the increased cross-shelf temperature gradient during austral winter. Fig. 9 shows the seasonal cycle of the zonal temperature gradient and SST as a function of distance to the coast. The seasonal cycle clearly reveals that the cooling and warming are not constant within 200 km distance to the coast throughout the year. The strongest negative zonal temperature gradient is

found between April and September with a secondary maximum in December/January. This increased gradient cannot be explained by the net surface heat flux. The difference between net surface heat fluxes in the coastal and the offshore boxes experiences its seasonal maximum in austral winter. Thus, the net surface heat flux act to damp the observed zonal SST gradient. The difference between the horizontal heat advection in the coastal and the offshore boxes is small. Furthermore, the seasonal cycle of this difference does not correspond to the seasonal changes of the zonal temperature gradient. Hence, the

mean horizontal advection likely plays no role for the increased zonal temperature gradient in austral winter.

Summarizing, stratification changes connected to the passage of CTWs, the seasonal cycle of SSS, and the changing net surface heat fluxes likely influence how effective the ML close to the coast is cooled by the dissipation of the internal tide, introducing a semi-annual cycle to the strength of the cross-shore temperature gradient. Nevertheless, the microstructure measurements suggests that the turbulent heat flux is an important cooling term throughout the year setting up the negative cross-shore

temperature gradient.



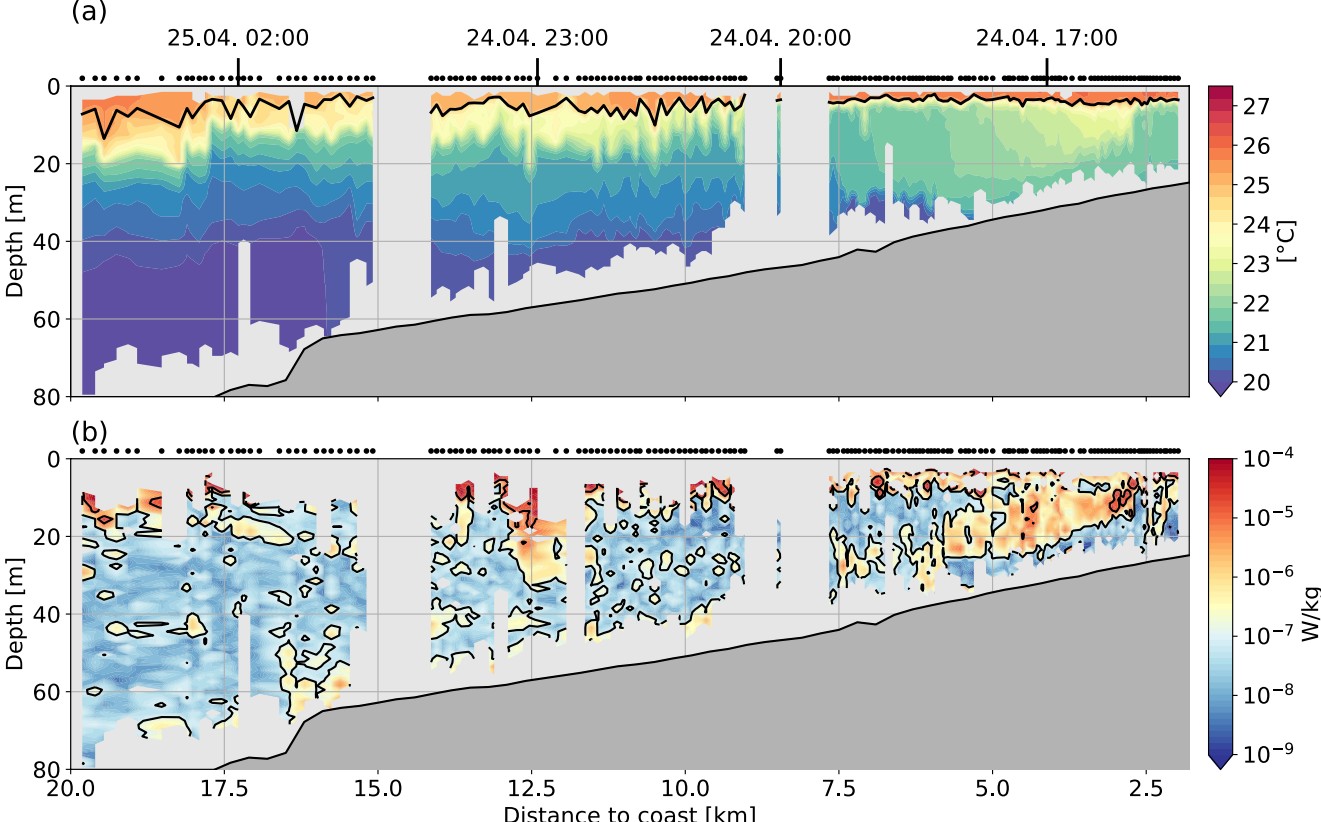

**Figure 10:** Shipboard section taken at 11°S in April 2022. (a) shows temperature (colours) and mixed layer depth (black line). (b) displays the dissipation rate of TKE below the mixed layer. Black points at the top of both panels mark the position of individual microstructure profiles.

The analysis and the turbulent heat flux calculated from the microstructure measurements revealed a high variability between different cruises. The data collected in April 2022 shows especially high fluxes (-390 W m$^{-2}$ in shallow waters, Fig. 10). The influence of turbulent mixing on the temperature field can be seen in the transect measured on the shelf of Angola in water depths between 25 m and 85 m in April 2022 (Fig. 10). The MLD decreased from around 7 m offshore to around 3 m towards the coast. The recorded section reveals strong internal wave activity as isotherms shows strong undulation indicative of onshore

propagating internal waves (Fig. 10a). This activity is primarily restricted to water depths larger than 50 m. In shallower water internal waves do not appear anymore suggesting breaking of internal waves and dissipation of internal wave energy.  It leads to high dissipation rates of TKE in this area with value mostly exceeding 10$^{-7}$ W kg$^{-1}$ here. The effect of the enhanced mixing due to breaking internal waves on the temperature field is pronounced. Temperatures are vertically much more homogenous near the coast than in deeper water. The high dissipation rates in this area are not directly connected to the ML as a local

minimum at around 10 m depth is detected. This suggests that the high mixing does not lead to a heat loss of the ML directly above. A very strong vertical temperature gradient of ~1 °C m$^{-1}$ below the ML supports the hypothesis that the mixing recorded



here mostly affects the layer below the ML. Consequently, non-local effects have to play a role. Note in this context that the averaged heat loss in this area is estimated to be 390 W m⁻² (Table 2). This heat loss is higher than the heat input by the net surface heat flux and the mean horizontal heat advection. It implies that a one-dimensional view is not sufficient to understand

turbulent heat loss at the Angolan shelf. Horizontal advection on small spatial and temporal scales likely plays an important role in the redistribution of heat. Note that the model results of Zeng et al. (2021) reveal a high spatial variability in dissipation at the Angolan shelf. This fits to our results and ultimately suggest that strong mixing and thus strong cooling in the tAUS locally occurs foremost in shallow and is redistributed by small scale horizontal advection. Processes that could be important in this context is the heat advection by nonlinear internal waves (Zhang et al., 2015) as well as the influence of eddy fluxes

(Thomsen et al., 2021). Further work has to be conducted to understand the redistribution of heat on small temporal and spatial scales in the tAUS.

Seemingly contradicting to our results, the study of Awo et al. (2022) showed that mean horizontal advection is an important term for the seasonal salinity budget. Their analysis shows that freshwater from the Congo River can reach 11°S by meridional advection in February/March and in October/November. Note that we also find a peak in southward surface velocities in

February and October. The velocities are also stronger in the northern tAUS until ~11°S (Fig. 3). However, as the meridional temperature gradient is weak in that region, the mean horizontal advection is not important for the local ML heat budget. Thus, the results of our study do not oppose the results found by Awo et al. (2022).

One shortcoming of the present study is the usage of the PREFCLIM climatology for the MLD. Using a climatology in combination with time series can introduce errors. As this issue concerns all terms discussed here, we argue that this issue does

affect all terms similarly. Another issue is that the minimum MLD in the PREFCLIM climatology is 10 m. Measurements, however, reveal that the MLD close to the coast can be shallower than 10 m (Fig. 10). Nevertheless, the small MLD are mostly found very close to the coast and thus affect only a small region. As the presented ML heat budget terms are averaged over the whole coastal box, we argue that the influence of this shortcoming is only minor.

The results of the present study shows that the residuum of the ML heat budget is likely explained by the turbulent heat loss

at the base of the ML and the uncertainties in the net surface heat flux primarily in the shortwave radiation. The uncertainties in the net surface heat flux represents a shortcoming of the study. This is especially important as the tAUS is a region with a large SST bias in state-of-the-art climate models (Richter 2015, Kurian et al. 2021, Farneti et al. 2022). One discussed reason for the bias is excessive shortwave radiation due to a poor representation of shallow stratocumulus clouds (Huang et al. 2007). Our results show that the uncertainty in shortwave radiation is seasonally dependent and higher in months when low level

clouds dominate (Scannell & McPhadden 2018). This indicates that the correct representation of clouds in both models and observations is still an issue. It has to be resolved, in order to get a better understanding of the tAUS and EBUS in general.

Summarizing, the study of the ML heat budget reveals that ML heat content changes in the tAUS are mostly determined by the surface heat fluxes and turbulent heat loss at the base of the ML. In contrast, the mean horizontal heat advection is of minor





importance. The surface heat fluxes determine the seasonal cycle of heating and cooling of the ML and act to damp the observed
cross-shore temperature gradient. Turbulent heat loss at the base of the ML acts throughout the year in shallow waters of the
tAUS. The microstructure data suggests that turbulent heat fluxes are capable of setting up the negative cross-shore temperature
gradient. Stratification changes seem to control the amount of turbulent heat loss at the base of the ML, introducing a semi-
annual cycle to the strength of the cross-shore temperature gradient.

**Appendix A**

**A.1 Comparison of satellite/reanalysis data to moored observation**

For the calculation of the ML heat budget, we rely on satellite/reanalysis data. To discuss uncertainties for the different dataset
we compare them to in situ measurements at the PIRATA-SEE mooring (6°S, 8°E). In the following, we will discuss
uncertainties based on the time series of the different variables (Fig. A1) as well as on the seasonal cycle (Fig. A2). Here, the
seasonal cycle of the satellite/reanalysis data is always calculated for the time period when the PIRATA-SEE data is available
for the individual variables and interpolated on the mooring location. Note that the incoming shortwave radiation in the
TropFlux product is multiplied by the factor 0.945 to account for the part of the radiation that is reflected at the sea surface
(Kumar et al., 2012). To compare the different datasets, we multiplied the shortwave radiation measured by the PIRATA-SEE
buoy and the MERRA2 data with the same factor.



**Figure A.1:** Time series of variables (see titles of subplots) of the ML heat budget at 6°S, 8°E from in-situ data collected at the PIRATA-SEE mooring (black line) and from satellite/reanalysis data (colours, see legend). The daily PIRATA-SEE data are interpolated on the same time grid as the satellite/reanalysis data.



**Figure A.2:** Seasonal cycle of variables (see titles of subplots) of the ML heat budget at 6°S, 8°E calculated from in-situ data
collected at the PIRATA-SEE mooring (black line) and from satellite/reanalysis data (colours, see legend). The seasonal cycle
from the satellite data is derived from the time period when PIRATA-SEE data is available.

### A.1.1 Surface heat fluxes



The surface heat fluxes are an important term of the ML heat budget in the tAUS as it is the largest warming term. From the time series differences between data from the in-situ fluxes measured by the PIRATA-SEE mooring and the surface fluxes from TropFlux and MERRA2 are recognizable (Fig. A1).

These differences are especially large for the shortwave radiation. Here, the shortwave radiation of TropFlux data always show higher fluxes than the PIRATA-SEE data. In contrast, the shortwave radiation from the MERRA2 product also exhibit lower heat fluxes than the in-situ data. These differences become even more evident looking at the seasonal cycles of shortwave radiation calculated from the different data products. The PIRATA-SEE data reveal a maximum in shortwave radiation in February and a minimum in August. The seasonal cycle of the TropFlux data also show a minimum in August. However, the maximum is found in January. The seasonal cycle also reveals that the TropFlux data records higher shortwave radiation throughout the year. A seasonal dependence of the differences is visible as they are smaller between March and July than during the rest of the year. The seasonal cycle of the MERRA2 shortwave radiation differ much more from the PIRATA-SEE data. The maximum is found in October and minimum in April. Between June and December, the shortwave radiation of MERRA2 is larger than the shortwave radiation of PIRATA-SEE and vice versa during the rest of the year.

The other terms of the surface heat fluxes show more agreement between the different datasets. The seasonal cycle of the longwave radiation calculated from TropFlux, MERRA2 and PIRATA-SEE data is similar. However, an offset between the different datasets exists. The seasonal cycle of longwave radiation calculated from the TropFlux (MERRA2) reveals less (more) radiation than in the seasonal cycle of the PIRATA-SEE data. For the latent heat flux the differences between MERRA2 and PIRATA-SEE are small ($\sim 3$ W m$^{-2}$). The differences between TropFlux and PIRATA-SEE are larger ($\sim 11$ W m$^{-2}$) with TropFlux showing less latent heat flux throughout the year. The contribution of the sensible heat flux to the net surface heat flux is in general small. Nevertheless, the seasonal cycle of MERRA2 shows a better agreement to the seasonal cycle of the PIRATA-SEE data than the TropFlux dataset.

After considering the results of the comparison between the satellite/reanalysis data and PIRATA-SEE data we decided to use the TropFlux dataset for shortwave and longwave radiation and MERRA2 for latent and sensible heat flux for the ML heat budget. We based this choice on the smallest root mean square (RMS) difference between the in-situ data and the different satellite/reanalysis products.

The comparison between the climatologies of the TropFlux/MERRA2 and the PIRATA-SEE data reveals a seasonal cycle in the differences between the different datasets. We want to investigate this bias by looking at the seasonal cycle of the mean differences between the time series (Fig. A3). The mean difference of shortwave radiation between the TropFlux product and the PIRATA-SEE measurements is higher than for the other heat flux terms and has a distinct seasonal cycle. Between July and February, the mean difference is higher ($\sim 15$ W m$^{-2}$) than from March to June ($\sim 5$ W m$^{-2}$). This seasonal cycle is most likely influenced by the seasonal prevalence of clouds of different types. Scannell and McPhaden (2018) show that at the PIRATA-SEE mooring site between January and April more high clouds then low clouds are present. This ratio is the other




way around during the rest of the year. Note that the shortwave radiation measured by the PIRATA-SEE mooring is on average

lower than what is measured by TropFlux suggesting that the satellite data overestimates the amount of shortwave radiation.

In contrast to the shortwave radiation, the latent and sensible heat flux provided by MERRA2 compare reasonably well with

the turbulent fluxes measured by the PIRATA-SEE buoy. Similarly, the mean difference of longwave radiation between

TropFlux and TropFlux data ranges between 1-8 W m$^{-2}$ throughout the year. The bias of the net surface heat flux is dominated

by the bias in shortwave radiation and ranges between 4 W m$^{-2}$ in April and 38 W m$^{-2}$ in January.

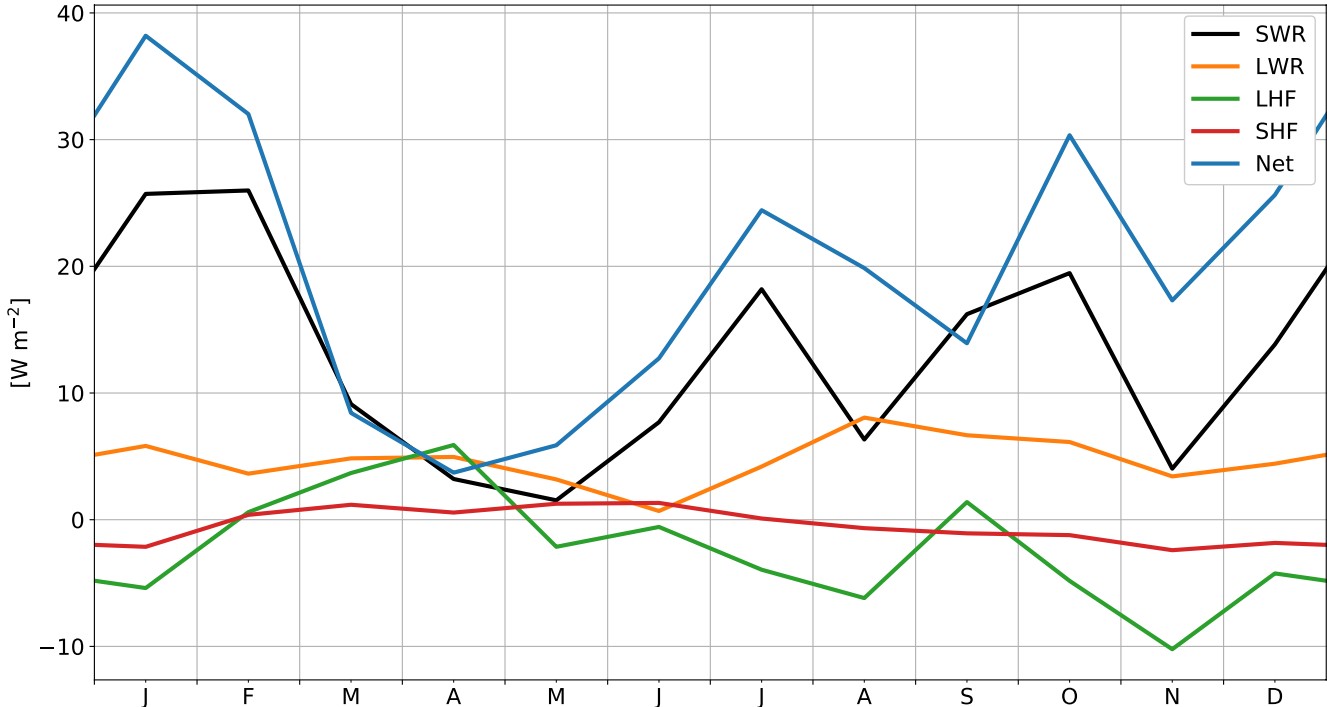

**Figure A3:** Climatology of the mean difference between the satellite/reanalysis data and the in-situ measurements at the PIRATA-SEE mooring. For the shortwave (SWR) and longwave radiation (LWR) the TropFlux is used, for latent (LHF) and sensible heat flux (SHF) the MERRA2 dataset is used.

Summarizing, comparisons of surface heat fluxes from different data sources show large uncertainties. Smallest differences

are achieved using TropFlux dataset for shortwave and longwave radiation and MERRA2 dataset for latent and sensible heat

flux.

To estimate the uncertainties of the sea surface heat fluxes for the ML heat budget we calculate the RMS differences between

the PIRATA-SEE and the satellite/ reanalyses data from all available months. This RMS difference of the shortwave radiation

is 20 W m$^{-2}$, for the longwave radiation 6 W m$^{-2}$, for the latent heat flux 10 W m$^{-2}$, for the sensible heat flux 2 W m$^{-2}$.

**A.1.2 Horizontal velocities**



We compare the horizontal velocities measured at the PIRATA-SEE mooring at 10 m depth of the OSCAR velocities. The seasonal cycle of meridional velocities is similar in both datasets. A minimum in southward velocity is found in March. During the rest of the year the velocities are small. The RMS difference based on monthly data is 7 cm s$^{-1}$. The seasonal cycle of the

zonal velocities shows an offset of around 7 cm s$^{-1}$. The RMS difference based on monthly data is 10 cm s$^{-1}$. Note that the zonal velocities show anomalous southward velocities during the end of the first mooring period (2007) that are much stronger than all other recorded data. Comparing only the latter mooring period from to the OSCAR data reveals a better agreement between both datasets (not shown).

### A.1.3 Surface temperatures

The comparison between surface temperatures measured by the PIRATA-SEE mooring and the OSTIA SST product shows a very good agreement. The RMS difference between both products based on monthly data is 0.1°C.

### Data availability

Publicly available datasets were used for this study. Data from TropFlux are from the Indian National Centre for Ocean Information Services and their website http://www.incois.gov.in/tropflux/. Data from MERRA2 are downloaded from their

website https://gmao.gsfc.nasa.gov/reanalysis/MERRA-2/. The OSTIA-SST were accessed via the Copernicus Server (https://marine.copernicus.eu). Surface velocities are from the OSCAR dataset (https://doi.org/10.5067/OSCAR-03D01). Mixed layer depths are taken from the PREFCLIM climatology (https://doi.org/10.1594/PANGAEA.868927). The data from the PIRATA South East Extension are available on the project website (https://www.pmel.noaa.gov/gtmba/).

### Author contribution

MK performed the analysis and drafted the paper. PB and MD led the observational programme at sea. All co-authors reviewed the manuscript and contributed to the scientific interpretation and discussion.

### Acknowledgements

The study was funded by EU H2020 under grant agreement 817578 TRIATLAS project. It was further supported by the German Federal Ministry of Education and Research as part of the SACUS (03G0837A), SACUS II (03F0751A), and

BANINO (03F0795A) projects and by the German Science foundation through several research cruises with RV Meteor. We would like to thank Volker Mohrholz for contributing to the microstructure measurements during part of the cruises. We thank the captains, crews, scientists, and technicians involved in several research cruises in the tropical Atlantic who contributed to collecting data used in this study. The TropFlux data is produced under a collaboration between Laboratoire d'Océanographie: Expérimentation et Approches Numériques (LOCEAN) from Institut Pierre Simon Laplace (IPSL, Paris, France) and National

Institute of Oceanography/CSIR (NIO, Goa, India), and supported by Institut de Recherche pour le Développement (IRD, France). TropFlux relies on data provided by the ECMWF Re-Analysis interim (ERA-I) and ISCCP projects.



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
