# Peer review of "Seasonal Cycle of Sea Surface Temperature in the Tropical Angolan Upwelling System"

_EGUsphere, 2022_

## Author Comment (AC1)

Authors response to comments by reviewer #1of the manuscript "Seasonal cycle of sea surface temperature in the tropical Angolan upwelling system" by Mareike Körner (mkoerner@geomar.de), Peter Brandt and Marcus Dengler.

We would like to thank the reviewer for the detailed and helpful comments to improve the manuscript. Below, we use black text for the reviewer's comments and blue text for our response.

The seasonal cycle of the mixed layer heat budget is analyzed in the southeastern tropical Atlantic near the African coast. There is a significant residual in the budget when comparing the rate of change of mixed layer heat content and the sum of the surface heat flux and horizontal oceanic heat advection. The residual is larger near the coast in the annual mean, suggesting a larger contribution from vertical turbulent cooling through the base of the mixed layer. Direct measurements of turbulence from several cross-shore cruises reveal stronger mixing and turbulent cooling in the near-shore region, consistent with the heat budget results. It is also hypothesized that seasonal variations in temperature stratification my generate a seasonal cycle in turbulent cooling that drives seasonal differences in the cross-shore SST gradient.

The manuscript is well-written and organized, though there are numerous minor edits that are needed to the language/grammar (see detailed comments below). The results are interesting and will be useful for understanding the mechanisms of mixing and SST variability in the Angolan region and possibly more generally in coastal areas, and for validating models since many have large biases in the southeastern tropical Atlantic. The conclusions are supported well by the analysis and results. I have a few main comments for the authors to consider during their revision.

Thank you for very much for the positive evaluation of our manuscript. Please find the detailed replies to your comments below.

**Main comments:**

Lines 301 and 381: the turbulent heat flux is averaged between 2 and 15 m below the ML: why use this depth range? Are results similar for smaller depth ranges? 15 m seems deep for mixing to affect SST. Is there any precedent for using this depth range?

Thank you, indeed, we did not motivate our choice of depth range in the previous version of the manuscript. The depth range 2m to 15m below the ML is a trade-off between regions where we are able to estimate turbulent eddy diffusivities from microstructure data and statistical reliability of our results. We will explain this in more detail here, but also included additional text in the manuscript motivating this choice. Previous studies that used similar depth ranges to estimate turbulent heat loss of the ML include Hummels et al. (2013; 2014) (5-15m below ML), Hummels et al. (2020) (1m-10m below ML) and Moum et al. (2013) who calculated a mean heat flux from values between 20m and 60m depth in the equatorial Pacific, but omitted values from within the ML.

Our aim is to determine the turbulent heat flux across the base of the ML. As the turbulent heat flux vanishes at the sea surface, its value across the ML base represents the turbulent heat flux divergence of the ML and thus the ML heat loss. However, we are facing two issues, the inability to accurately determine eddy diffusivities in regions of low or vanishing stratification, and the need for many statistically independent estimates of turbulent dissipation rates to gain statistical confidence in our results.

The first issue is related to the uncertainty of mixing efficiency  $\Gamma$  in low or unstratified environments, as in the ML.  $\Gamma$  is needed to infer turbulent eddy diffusivities from dissipation rates of turbulent kinetic energy (TKE) (see Equation 4 of manuscript). While it has been shown

in several studies that  $\Gamma$ =0.2 is a good approximation in most oceanic regions where turbulence and stratification is elevated (recently summarized by Gregg et al. 2018), this approach is not valid in regions of low or vanishing stratification.  $\Gamma$ , defined as the ratio of change of background potential energy and expended energy, must greatly decrease in regions of nearvanishing stratification, but it is unclear how to parametrize this adequately (e.g. Gregg et al., 2018). Thus, to spare from making inadequate assumptions, we refrain from estimating turbulent eddy diffusivities and thus turbulent heat flux in the ML and across the mixed layer base where stratification is low.

The other issue is related to statistical uncertainty. Dissipation rates of TKE follow a nearlognormal distribution (e.g. Davis, 1996) spanning 5 orders of magnitude in our data set. Strongly elevated values occur very rarely but to a large extend determine the magnitude of the heat flux. It is this suggested that heat flux estimates should be carried out with large data ensembles. Our data processing allows to determine a statistical independent estimate of the dissipation rate of TKE about every 1.2m in the water column. Due to the limited number of microstructure profiles available, we found that including samples from 2m to 15m below the ML was an optimal choice to obtain reasonable confidence limits of our heat flux estimates. Certainly, as the turbulent heat flux is (mostly) convergent with increasing depth below the ML, our estimates are likely biased low.

In the method section 3.1.3. we now state (lines 219 - 220):

"All measurements in the ML as well as 2 m below are disregarded because mixing efficiency  $\Gamma$  is unknown in low-stratified waters (Gregg et al., 2018) and to avoid using data impacted by ship turbulence. "

And two sentences later we state (lines 223 – 227):

"To evaluate the ML heat loss, all individual estimates in the depth range between 2 m and 15 m below the ML are averaged. The use of this depth range is a trade-off between regions where we are able to estimate turbulent eddy diffusivities from microstructure data and statistical reliability of our results. Due to the lognormal distribution of the dissipation rates of TKE (e.g., Davis, 1996), it is desired to average many individual estimates of  $\epsilon$  to increase statistical reliability. "

As the authors mention, the use of climatological MLD introduces uncertainty. What are the expected errors introduced by the MLD climatology? How do they affect the uncertainties in the heat budget terms? Can you estimate them by comparing the clim. MLD that you used to the actual MLD calculated from the PIRATA mooring? With such a thin climatological ML, small errors could have a big impact on the magnitudes/errors of the heat budget terms and error bars on residual.

Thank you for this comment. After some further analyses we decided to use a different dataset for the MLD in the revised version. The main reason for this was our finding that the MLD of the PREFACE climatology were biased high. This is likely due to the fact that the climatology used a MLD minimum of 10 m for all individual CTD profiles that went into the climatology. Particularly in region of the continental margin, MLDs are often smaller.

In the revised manuscript we now make use of the GLORYS reanalysis product. The GLORYS reanalysis product has a higher spatial resolution and is available daily within the time span of the study (1993-2018). It compares well to the MLD at the PIRATA-SEE mooring location (Fig. R1). Furthermore, it reasonably well reproduces the shallow MLD at the coast in the tAUS found in the Nansen Data (Fig. R2).

Making use of another dataset for the MLD, changes the magnitude of the heat budget terms (Fig. R3). Due to the decreased MLD in the GLORYS product, all evaluated terms of the ML heat budget decreased so that the net effect on the residual of the heat budget is small.

**Figure R1:** Climatology of the mixed layer depth at the PIRATA-SEE mooring site (6°S, 8°E) calculated with the PIRATA mooring data (green line), the GLORYS reanalysis product (blue line), and the PREFCLIM climatology (orange line).

---

## Author Comment (AC2)

Authors response to comments by reviewer #2 of the manuscript "Seasonal cycle of sea surface temperature in the tropical Angolan upwelling system" by Mareike Körner (mkoerner@geomar.de), Peter Brandt and Marcus Dengler

We would like to thank the reviewer for the detailed and helpful comments to improve the manuscript. Below, we use black text for the reviewer's comments and blue text for our response.

**Summary**

This study focusses on understanding the seasonal cycle of Sea Surface Temperature (SST) in the tropical Angolan Upwelling System and in particular look at the processes responsible for an increased cross-shore SST gradient in winter. To do so, the authors analyse the seasonal mixed layer heat budget using satellites and reanalysis datasets in two boxes: at the coast and offshore. By comparing the temperature rate of change to the surface heat fluxes, horizontal advection and residual, they found (1) a strong contribution of the surface heat fluxes that sets the seasonal cycle counterbalance by (2) a significant contribution of the residual; (3) The contribution of the horizontal advection is shown to be minor. Interestingly, the cooling contribution of the residual is larger near the coast than offshore suggesting a key role of the residual in driving the increased SST crossshore gradient in winter. Turbulent heat fluxes estimated from shipboard measurements show a strong cooling effect of turbulent mixing, which is particularly strong at the coast. This process might explain the increased cross-shore SST gradient.

This is an important topic, and the paper is well-structured and clear. I do have reservations about the large uncertainties that might exist due to the use of various datasets, the gaps between data resolutions and the contribution of the other processes included in the residual but the major points are well discussed in the discussion and the main conclusions of the paper are still valid. There are a few general remarks and minor corrections of the text that I feel that the authors should address, but overall, it is a substantive piece of work of good quality and worthy of publication.

Thank you very much for your thorough examination and positive evaluation of our manuscript. Please find the detailed answers to your comments below.

**General Remarks**

Introduction: L30-35. When reading the introduction, I will have found it nice to see a plot with the seasonal cross-shore SST gradient and primary productivity (the paper's main motivation). Maybe, as a last subplot of Fig. 1?

Thank you for this comment. We added a second figure to the introduction showing the seasonal cycle of SST and net primary production averaged over the coastal and offshore boxes.

Introduction: L45 to 61. In this part of the introduction, you describe the SST seasonal cycle, emphasizing that the latter cannot be explained by the upwelling-favourable winds. What about the net heat flux that I guess is a major contributor to the seasonal cycle SST in the tAUS?

Yes, the net surface heat fluxes strongly influence the seasonal cycle of SST. What we want to explain in this paragraph is that other eastern boundary upwelling systems are also characterized by colder waters near the coast and high productivity. These signals can be associated with local wind-driven upwelling. In the tAUS the wind cannot explain the colder waters and enhanced productivity near the coast as the winds are weak and the seasonal cycle of neither the alongshore wind stress nor of the wind stress curl fit to the seasonality of these signals.

**We clarified this paragraph:**

Line 50-55: "In contrast to other eastern boundary upwelling systems (EBUS) the changes in surface temperatures, the cross-shelf temperature gradient, and the productivity signal in the tAUS cannot be explained by local wind-driven upwelling (Ostrowski et al., 2009). On average, the winds in tAUS are southwesterly and substantially weaker than in the Benguela upwelling system (Fig. 1b). They have a weak seasonal cycle with a minimum in upwelling-favourable winds during the upwelling season in austral winter (Ostrowski et al., 2009; Zeng et al., 2021)."

L160-163: Why did you decide to interpolate variables with coarser resolution onto a higher resolution grid? Usually, the inverse is done to prevent the generation of fake information. How much do you think this might affect the contribution of mean horizontal advection?

We decided to interpolate the dataset with the coarser resolution linearly onto the higher resolution grid to make sure that we do not loose information. We tested the sensitivity of our results to the interpolation scheme by interpolating the data onto the coarsest available grid (Fig. R1).

**Figure R1:** (a) & (b) Individual contributions to the ML heat budget. Colors are explained in the legend. (c) & (d) Sum of net surface heat flux and horizontal heat advection (green lines), the observed heat content change (black lines), and the resulting residuum between both (red dashed line). (a) & (c) show the result averaged over the offshore box, (b) & (d) display the results averaged over the coastal box. Solid lines give the results by interpolating on the grid with the highest resolution for each term. Dashed lines give the results interpolating on the coarsest grid for each term.

The results from both interpolation schemes only show small differences that are slightly elevated in the coastal box. Note that in the revised version of our manuscript, we use a different MLD climatology. MLD are now calculated from the GLORYS reanalyses product which allows an improved representation of the shallow MLD close to the coast. The GLORYS MLD are available on a 1/12° grid. The spatial variability of MLD is most likely the main contributor to the differences of the ML heat budget terms of both interpolation schemes. Overall, we argue that the differences due to interpolation are small and that the general results

are independent of the grid that is interpolated on. In the revised version of the manuscript, we thus decided to retain interpolating data on the grid with the higher resolution.

L254 – It is interesting that the SW is higher in the coastal box than in the offshore box. I will have thought the contrary due to the important cloud cover along the coast. Do you think that the bias in the SW could be more important along the coast?

The differences in incoming SWR between both boxes differ throughout the year. Between May and October, the incoming SWR in the coastal box is higher. The period between May and October corresponds to the season where low clouds are dominantly found over the south eastern tropical Atlantic (Scannell & McPhaden 2018). The cloud cover is stronger further offshore than directly at the coast (Zuidema et al. 2016). Thus, more incoming SWR in the coastal box than in the offshore box can be expected during this period.

In the previous version of the manuscript, we just showed and discussed the amount of SWR that is absorbed in the ML. We realized that it might be of interest to also show the seasonal cycle of incoming SWR. In the revised version this is added to Figure 5.

We changed the paragraph accordingly:

Line 262-270: "In both boxes, the incoming shortwave radiation peaks in January/February. The minimum in July is driven by the seasonal maximum in solar zenith angle and the expansion of the low cloud cover (Scannell and McPhaden, 2018). The cloud cover is stronger further away from the coast (Zuidema et al., 2016) leading to higher incoming shortwave radiations over the coastal box. For the analyses of the ML heat budget, the amount of shortwave radiation that is absorbed within the ML is of interest. The fraction of shortwave radiation that is absorbed in the ML depends on the MLD and the chlorophyll concentration (see section 3.1.1). This fraction is largest in austral winter when the MLD and Chlorophyll concentration absorbed in the ML is higher in the coastal box. The largest differences are observed in July when the ML in the coastal box receives 17 W m-2 more shortwave radiation."

Figure8: From the figure, it seems that the rate of change in winter is more negative offshore than inshore (except for April maybe). Should not be the cooling stronger at the coast, resulting in an increased SST cross-shore gradient in winter. Could you provide a plot showing the difference of heat content change in the two boxes?

Yes, the heat content change is more negative in the offshore box than in the coastal box in austral winter (Fig. R2 a). The change in heat content depends on the change of SST over time as well as on the mixed layer depth. The mixed layer depth is deeper in the offshore box than in the coastal box (Fig. 4 in the revised manuscript). If we consider just the seasonal cycle of the change of temperature, we do see that the change in austral winter is more negative in the coastal box in April and of the same order from May-July (Fig. R2 b).

One other general thing to consider is that the difference in temperature between the coastal and offshore box is not a perfect proxy for the zonal SST gradient as the position of the SST maximum varies throughout the year (see Fig. 10 in the manuscript).

In the revised manuscript we now mention these points:

Line 383 - 386: "The heat content change reveals that the ML cools from March to August in both boxes. In austral winter the change is stronger in the offshore box than in the coastal box. This may seem counterintuitive given the increased negative cross-shore SST gradient during this period, but is explained by the deeper ML in the offshore box compared to the coastal box (Figs. 4c, f)."

Line 463 - 467: Fig. 10 shows the seasonal cycle of the zonal temperature gradient and SST as a function of distance to the coast. It clearly reveals that the cooling and warming are not

constant within 200 km distance to the coast throughout the year. Particularly, the zonal maximum of SST (zero contour line of the zonal temperature gradient, Fig. 10) is in some months within the coastal box and in other months within the offshore box. Note that temperature differences averaged over both boxes (Fig. 2) are thus not a perfect proxy for the cross-shelf temperature gradient."